# END-TO-END TRAINING OF UNSUPERVISED TREES: KAURI AND DOUGLAS

## ABSTRACT

Trees are convenient models for obtaining explainable predictions on relatively small datasets. While many proposals exist for end-to-end construction of such trees in supervised learning, learning a tree end-to-end for clustering without labels remains an open challenge. As most works focus on interpreting with trees the result of another clustering algorithm, we present here two novel end-to-end trained unsupervised trees for clustering, respectively Kauri for datasets with a large number of features using binary decision trees, and Douglas for datasets with a large number of samples using $k$-ary differentiable trees. Both methods are composed of a learnable tree structure in which parameters are optimised according to a generalised mutual information (GEMINI) and present results on par with other existing methods while maintaining interpretability. We compare these two models on multiple datasets with the most recent unsupervised trees and provide guidelines for choosing the most suitable model.

## 1 INTRODUCTION

Decision tree classifiers are one of the most intuitive models in machine learning owing to their intrinsic interpretability (Molnar, 2020, Section 3.2). Trees consist of a set of hierarchically sorted nodes starting from one single root node. Each node comprises two or more conditions called rules, each of which leading to a different child node. Once a node does not have any child, a decision is returned. A childless node is named a leaf.

While the end model is eventually interpretable, building it implies some questions to be addressed, notably regarding the number of nodes, the feature (or set of features) on which to apply a decision rule, the construction of a decision rule i.e. the number of thresholds and hence the number of children per node. Learning the structure is easier in the case of supervised learning, whereas the absence of labels makes the construction of unsupervised trees more challenging. In recent related works, the problem was oftentimes addressed with twofold methods (Tavallali et al., 2021; Laber et al., 2023): first learning clusters using another algorithm e.g. KMeans, then applying a supervised decision tree to uncover explanations of the clusters. However, such *unsupervised trees* are not fully unsupervised in fact since their training still requires the presence of external labels for guidance which are provided by KMeans.

To achieve end-to-end unsupervised learning in trees, we propose a framework where we merge the view of trees as statistical models with learnable parameters and a clustering criterion to maximise: the generalised mutual information (GEMINI, Ohl et al., 2022), a distance-based score. We derive two new clustering algorithms from this framework; respectively binary decision trees for datasets with a large number of features (Kauri) and $k$-ary differentiable trees for datasets with a large number of samples (Douglas). A short description of these methods is provided in Fig. 1. The contributions of this framework are therefore:

- The introduction of two end-to-end unsupervised trees for clustering: Kauri and Douglas. Both approaches learn a tree architecture using GEMINI maximisation. The former uses binary decision trees and the latter differentiable trees.
- We show that Kauri displays equal performance in clustering to kernel KMeans+Tree using end-to-end training while obtaining shallower structures.
- A practical example showing how to interpret the obtained models in clustering

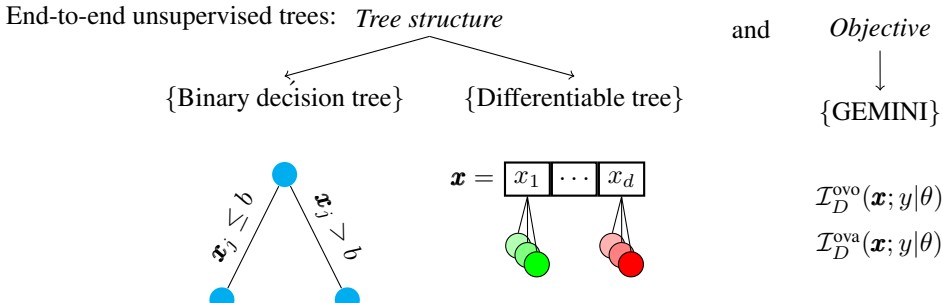

Figure 1: Summary of the proposed framework for learning end-to-end unsupervised trees. The framework concatenates a tree structure with an objective to maximise: the generalised mutual information. The Kauri model corresponds to a binary decision tree with the squared-MMD GEMINI whereas the Douglas model corresponds to a differentiable tree and the Wasserstein GEMINI.

## 2 Training trees

We progressively present in this section the different means for creating a decision tree structure, with supervision or not and linking the algorithm with discriminative and hierarchical clustering methods.

### 2.1 How do we train supervised trees?

In supervised learning, we have access to targets $y$ which guides our tree construction for separating well our samples. In this field, we can refer to the well-known classification and regression tree (CART) (Breiman et al., 1984). At each node, we evaluate the quality of a split, i.e. a proposed rule on a given feature and data-dependent threshold, through gain metrics. We then add to the tree structure the split that achieved the highest possible gain. Common implementations of supervised trees use the Gini criterion developed by the statistician Corrado Gini (1912), which indicates how *pure* a tree node is given the proportion of different labels in its samples (Casquilho & Österreicher, 2018). Later works then proposed different gain metrics like the difference of mutual information in the ID3 (Quinlan, 1986) and C4.5 (Quinlan, 2014) algorithms.

When the number of leaves is unlimited, these approaches can produce deterministic outputs. Moreover, their greedy nature can lead to the construction of very deep trees which harms the interpretable nature of the model (Luštrek et al., 2016). This motivates for example the construction of multiple trees that are equivalent in terms of decision, yet different in terms of structure presenting thus an overview of the Rashomon set for interpretations (Xin et al., 2022). Other approaches tried to overcome the deterministic non-differentiable nature of the rule-based tree by introducing differentiable leaves (Fang et al., 1991; Yang et al., 2018) which allows to train trees through gradient descent. We will later come back to the definition of one such model for our method, the deep neural decision tree (Yang et al., 2018).

Whether differentiable or not, we choose to describe the decision trees as statistical models $p_\theta(y|\boldsymbol{x})$ which assign the data sample $\boldsymbol{x}$ to a discrete variable $y$, the cluster membership, according to some parameters $\theta$. These parameters can be for example the set of thresholds and features on which decisions are carried at each node or matrix weights in differentiable trees as we will see in the next sections.

### 2.2 How do we train unsupervised trees?

In clustering, we do not have access to labels making all previous notions of gains unusable so we need other tools for guiding the splitting procedure of the decision trees. A common approach is then to keep the algorithm supervised as described in the previous section, yet providing labels that were derived from a clustering algorithm e.g. KMeans (Laber et al., 2023; Held & Buhmann, 1997). In this sense, derived centroids from KMeans can be involved as well in splitting procedures (Tavallali et al., 2021), even to the point of not needing the data from which the centroids

are derived (Gamlath et al., 2021). However, such methods do not properly construct the tree *from scratch* in an unsupervised way despite potential changes in the gain formulations. We are interested in a method that can provide a directly integrated objective to optimise for tree training. For example, Bertsimas et al. (2021) directly optimise the silhouette score, an internal clustering metric, yet report the need for warm start to train multivariate decision trees. Other gains derived from entropy formulations can also be proposed (Bock, 1994; Basak & Krishnapuram, 2005). We even note the usage of recursive writing of the mutual information to achieve deeper and deeper refinements of binary clusters (Karakos et al., 2005). Oftentimes, these approaches assume that a leaf describes fully a cluster. Combining leaves into a single cluster requires then post-hoc methods (Fraiman et al., 2013).

If we allow post-hoc methods, an elegant approach to constructing an unsupervised tree was proposed by Liu et al. (2000) by adding uniform noise to data and tasking a decision tree to separate noise from true data. Such trees put in different leaves dense areas of the data which can then be labelled manually for example.

### 2.3 Generalised mutual information for clustering

Inspired by the involvement of mutual information in tree gains for clustering (Karakos et al., 2005), we are interested in finding an easy-to-compute gain that requires no model-based hypotheses on the data and which does not involve a first-stage clustering algorithm for guidance.

The generalised mutual information (GEMINI) (Ohl et al., 2022) is a cost function introduced to perform clustering with any discriminative model taking the form $p_\theta(y = k|\boldsymbol{x})$ linking the discrete cluster assignment $y$ to the data $\boldsymbol{x}$ through parameters $\theta$. Maximising this loss implies maximising a statistical distance $D$ between the cluster distribution $p_\theta(\boldsymbol{x}|y = k)$ among randomly chosen clusters. While defined on the distributions $p_\theta(\boldsymbol{x}|y = k)$, Bayes theorem leads to a computable formula of this loss function involving only the prediction of the model $p_\theta(y = k|\boldsymbol{x})$. Contrary to most recent unsupervised losses, especially contrastive losses, the GEMINI requires neither regularisations nor data augmentation to achieve clustering. Its most defining input is a well-chosen metric in the data space which can be a kernel if the statistical distance $D$ is the maximum mean discrepancy (MMD) (Gretton et al., 2012) or a distance if the statistical distance is the Wasserstein (Peyré & Cuturi, 2019). GEMINI has two definitions; the one-vs-all:

$$\mathcal{I}_D^{\text{ova}}(\boldsymbol{x}; y|\theta) = \mathbb{E}_{y \sim p_\theta(y)}[D(p_\theta(\boldsymbol{x}|y)\|p(\boldsymbol{x}))], \tag{1}$$

and the one-vs-one:

$$\mathcal{I}_D^{\text{ovo}}(\boldsymbol{x}; y|\theta) = \mathbb{E}_{y_a, y_b \sim p_\theta(y)}[D(p_\theta(\boldsymbol{x}|y_a)\|p_\theta(\boldsymbol{x}|y_b))]. \tag{2}$$

This metric was originally intended for gradient descent methods, especially neural networks. However, we will show here how we can revisit the GEMINI for tree models which can be non-differentiable, leveraging end-to-end learning.

## 3 KAURI: KMEANS AS UNSUPERVISED REWARD IDEAL

The Kauri tree is a non-differentiable binary decision tree that looks in many ways alike the CART algorithm. It constructs from scratch a binary tree giving hard clustering assignments to the data by using an objective equivalent to both the optimisation of a kernel KMeans and an MMD-GEMINI. In the Kauri structure, a cluster can be described by several leaves.

### 3.1 Notations and modelling

We consider that we have a dataset of $n$ samples: $\mathcal{D} = \{\boldsymbol{x}_i\}_{i=1}^n$. We can model the classification/clustering distribution associated with decision trees as a delta Dirac:

$$p_\theta(y = k|\boldsymbol{x}) = \mathbb{1}[\boldsymbol{x} \in \mathcal{X}_k], \tag{3}$$

with $\{\mathcal{X}_k\}_{k=1}^K$ a partition of the data space $\mathcal{X}$. Notice that we use the notation $\mathbb{1}$ because $y$ is discrete. We set $\mathcal{X} \subseteq \mathbb{R}^d$. We write the partition into $K$ clusters as the sets of the indices of the samples that fall in the respective data subspace:

$$\mathcal{C}_k = \{i|\boldsymbol{x}_i \in \mathcal{X}_k\}, \forall k \leq K. \tag{4}$$

We assume that the model sees all the data and that $p(x)$ corresponds to the empirical distribution of the training data. Consequently, we do not use minibatches and write the expectations of the model turn to discrete sums. Notably, we have:

$$p_\theta(y = k) = \frac{|\mathcal{C}_k|}{n}. \tag{5}$$

We note $\mathcal{N}_p$ the set of samples reaching the $p$-th node and $b^p$ its threshold defined for a single feature $j$. This threshold defines two binnings and produces two child nodes. For example, if $\boldsymbol{x}^j \leq b^p$, then this sample goes to the left child of the parent node $p$, otherwise to the right child.

## 3.2 TREE BRANCHING

For supervised trees like CART or ID3, the types of splits are binary and guided by the labels which tell us to which class each child node should go. For unsupervised trees, we must consider all possibilities: to which cluster goes the left child, to which cluster goes the right child on which feature to do the split, on what threshold in this feature to split, on which nodes. Assuming to be located at a node $p$ for a split, let $\mathcal{S}_L$ the subset of samples from the node samples $\mathcal{N}_p$ that will go to the left child node and $\mathcal{S}_R$ the complementary subset of samples that will go to the right child node. Each child node will be assigned to a different cluster, whether new, already existing or equal to the parent node's cluster assignment. Let $k_p$ be the current cluster membership of the parent node $p$, $k_L$ the future cluster membership for the left child node and $k_R$ the future cluster membership of the right child node, i.e. $\mathcal{S}_L \cup \mathcal{S}_R = \mathcal{N}_p \subseteq \mathcal{C}_{k_p}$ and after splitting: $\mathcal{S}_L \subseteq \mathcal{C}_{k_L}$ and $\mathcal{S}_R \subseteq \mathcal{C}_{k_R}$.

We enforce the following constraints: a child node must stay in the parent node's cluster if both children leaving would empty the parent's cluster; the creation of a new cluster can only be done under the condition that the number of clusters does not exceed a specified limit $K_{\max}$. We also impose a maximum number of leaves $L_{\max}$ which can be equal to at most the number of samples $n$. It is nonetheless possible that the algorithm stops the splitting procedure if all gains become negative before reaching the maximum number of leaves allowed.

Thus, learning consists in greedily exploring from all nodes the best split and either taking this split to build a new cluster or merging with another cluster. We now present the objective function and related gains depending on the children's cluster memberships.

## 3.3 GAIN METRICS

Kauri is designed to maximise the following objective function:

$$\mathcal{L} = \sum_{k=1}^{K_{\max}} \frac{\sigma(\mathcal{C}_k^2)}{|\mathcal{C}_k|}, \tag{6}$$

where the function $\sigma$ sums the kernel values $\kappa = \langle \varphi(\boldsymbol{x}_i), \varphi(\boldsymbol{x}_j) \rangle$ of samples indexed by two sets:

$$\sigma(E, F) = \sum_{\substack{i \in E \\ j \in F}} \kappa(\boldsymbol{x}_i, \boldsymbol{x}_j), \tag{7}$$

We will refer to the $\sigma$ function as the *kernel stock*. This function is bilinear with respect to the input spaces. The objective in Eq. 6 corresponds simultaneously to the maximisation of one-vs-all or one-vs-one squared MMD GEMINI or the minimisation of a kernel KMeans objective. The proofs are provided in App. B. We can derive from this objective four gains that evaluate how much score we get by assigning one child node to a new cluster, assigning both child nodes to two new clusters, merging one child node to another cluster or merging both child nodes to different clusters. We denote by $\mathcal{C}'_\bullet$ the clusters after the split operation and $\mathcal{C}_\bullet$ the clusters before the split. Hence, the global gain metric is:

$$\Delta\mathcal{L}(\mathcal{S}_L : k_p \to k_L, \mathcal{S}_R : k_p \to k_R) = \frac{\sigma(\mathcal{C'}_{k_L}^2)}{|\mathcal{C}'_{k_L}|} + \frac{\sigma(\mathcal{C'}_{k_R}^2)}{|\mathcal{C}'_{k_R}|} + \frac{\sigma(\mathcal{C'}_{k_p}^2)}{|\mathcal{C}'_{k_p}|}$$
$$- \frac{\sigma(\mathcal{C}_{k_L}^2)}{|\mathcal{C}_{k_L}|} - \frac{\sigma(\mathcal{C}_{k_R}^2)}{|\mathcal{C}_{k_R}|} - \frac{\sigma(\mathcal{C}_{k_p}^2)}{|\mathcal{C}_{k_p}|}, \tag{8}$$

Table 1: Advantages and disadvantages of the Kauri and Douglas algorithms for unsupervised tree construction.

|  | Splits | Scalable with $n$ | Scalable with $d$ | Hyperparameters |
|---|---|---|---|---|
| Kauri | Binary | No | Yes | $K_{\max}, L_{\max}$ |
| Douglas | $k$-ary | Yes with minibatches | No | Number of cut-points $T$ |

which corresponds to subtracting the contribution of the kernel stocks of the former clusters and adding the kernel stocks of the new clusters after splitting. From this global gain metric, we derive four different gains: the *star gain* $\Delta\mathcal{L}^{\star}$ for assigning either the left or right child of a leaf to a new cluster, the *double star gain* $\Delta\mathcal{L}^{\star\star}$ for assigning the left and right children of a leaf to two new clusters, the *switch gain* $\Delta\mathcal{L}^{\rightleftarrows}$ for assigning either the left or right child of a leaf to another existing cluster and the *reallocation gain* $\Delta\mathcal{L}^{\hookrightarrow}$ for assigning respectively the left and right children to different existing clusters. The algorithm can be bound in App. D, with an extended explanation of the derivations of the gains in App. C.

## 4 DOUGLAS: DNDTs OPTIMISED USING GEMINI LEVERAGE APPRISED SPLITS

The Douglas model seeks the full potential of GEMINI by combining it with differentiable trees. Thanks to this choice of architecture, we can optimise the Wasserstein-GEMINI, an objective more efficient for clustering than the MMD-GEMINI, with respect to the parameters through gradient descent. Indeed, the MMD-GEMINI only carries information through the means of cluster distributions and does not encompass all information on the data space whereas the expected Wasserstein distance between two randomly chosen clusters will take into account the complete distribution. However, the cost of Douglas is the loss of depth in tree as all rules are produced at the root level.

Deep neural decision trees (DNDTs, Yang et al., 2018) aim at learning individual rules per feature and then merge those rules to provide a final decision. Formally, each feature $f$ among a subset of selected features is assigned a vector of sorted thresholds $b_{1\cdots T}^{f}$ that determines the binnings of the feature. By defining a bias $\boldsymbol{c}^{f} = [0, -b_{1}^{f}, -b_{1}^{f} - b_{2}^{f}, \cdots, -b_{1}^{f} - b_{2}^{f} - \cdots - b_{T}^{f}]$ and a vector $\boldsymbol{a}^{f} = [0, 1, \cdots, T]$, Yang et al. (2018) write a feature-wise probability distribution with:

$$p_{\boldsymbol{a}^f, \boldsymbol{c}^f}(\beta|\boldsymbol{x}^f) = \text{SoftMax}\left(\frac{\boldsymbol{a}^f \boldsymbol{x}^f + \boldsymbol{c}^f}{\tau}\right), \tag{9}$$

named soft-binning where $\tau$ is a temperature hyperparameter set to 0.1. After each individual soft binning is applied, all combinations of features are computed using a Kronecker product, making DNDTs hardly scalable in terms of features. For example, if the $d$ features are all separated in $T+1$ binnings, the final decision will contain $(T+1)^d$ entries per sample. To produce a decision from this entry, a matrix multiplication with some parameters $\boldsymbol{W}$ is applied. The global model can be described as:

$$p_{\theta}(y = k|\boldsymbol{x}) = \sum_{t_1=1}^{T} \sum_{t_2=1}^{T} \cdots \sum_{t_d=1}^{T} W_{k, t_1 + dt_2 + \cdots + d^{d-1}t_d} \prod_{f=1}^{d} p_{\boldsymbol{a}^f, \boldsymbol{c}^f}(\beta = t_f|\boldsymbol{x}^f). \tag{10}$$

This model is therefore differentiable and can be trained by gradient descent.

For interpretation purposes, we choose to exploit active cut points as proposed by Yang et al. (2018). This is the number of features for which the respective cut points parameters do not lie outside of the feature boundaries in the dataset. For example, if for a single cut value (two bins) the bias is lower or greater than all samples on its respective feature, then this cut point is not active and does not participate in the decision.

## 5 EXPERIMENTS

We start by proposing a summary of the advantages and limitations of both tree algorithms in Table 1. Overall, Kauri is recommended for small-scale datasets whereas Douglas can be used with

Table 2: Summary of the datasets used in the experiments. *The number of features may be slightly larger than the actual number of variables as discrete variables were one-hot encoded.

| Name | Avila | Breast cancer | Car evaluation* | US Congress | Digits |
|---|---|---|---|---|---|
| Samples | 20,867 | 683 | 1,728 | 435 | 1,797 |
| Features | 10 | 9 | 21 | 16 | 64 |
| Classes | 12 | 2 | 4 | 2 | 10 |

| Name | Haberman survival | Iris | Mice protein | Poker hand | Vowel |
|---|---|---|---|---|---|
| Samples | 306 | 150 | 552 | 990 | 1,025,010 |
| Features | 3 | 4 | 77 | 10 | 10 |
| Classes | 2 | 3 | 8 | 2 | 10 |

| Name | Wine |
|---|---|
| Samples | 178 |
| Features | 13 |
| Classes | 3 |

large datasets on condition that there are few features. It is important to note that Kauri, Douglas and KMeans-based related works are distance-based clustering algorithms. Consequently, these algorithms are sensitive to the scaling of the data, unlike supervised trees. Therefore, we will scale most of our datasets with standard scaling to avoid the overtaking of specific features against all others due to large ranges. The summary of these datasets can be found in Table 2. For the sake of simplicity, we discarded most dataset samples with missing values unless specified otherwise. We will assess the general clustering performances and explanation power of the models before showing qualitative examples of their interpretation. Extended experiments can be found in App. H for an extended benchmark, App. I for model selection and App. G for an alternative version of Douglas.

## 5.1 ON THE CLUSTERING PERFORMANCES

We compare the performances of our two proposal algorithms on 10 datasets against recent methods for unsupervised tree constructions, namely ExShallow and RDM by Laber et al. (2023), and IMM (Moshkovitz et al., 2020). These methods are twofold and start by fitting KMeans centroids to the data, then learning a tree to explain the obtained clusters. The differences in all methods lie in attempts to limit the depth of the tree for the sake of simple explanations as deep trees tend to lose expressivity in explanation. We choose to provide a combination of KMeans and a standard CART decision tree classifier as a baseline for Kauri which is a kernel-KMeans-aimed clustering algorithm. For the twofold algorithms, we report the clustering performances according to the tree. As related works focus on trees with one leaf per cluster, we limit the Kauri tree and the KMeans+Tree to as many leaves as clusters. For results regarding more leaves than clusters and a comparison with related work ExKMC by Frost et al. (2020), please refer to App. H where the results remain consistent.

Since some algorithms are deterministic in nature, we introduce stochasticity in results by selecting 80% of the training data over 30 runs. Details on preprocessing and experimental hyperparameters are reported in App. E. We report the performances in terms of adjusted rand index (ARI, Hubert & Arabie, 1985), a common clustering external metric, for all algorithms in Table 3 and in terms of KMeans score normalised by the actual KMeans performance (Laber et al., 2023) in Table 4. Due to scores being all equal to 0, we discarded the Poker hand dataset fomr Table 3. As mentioned before, Douglas' complexity grows exponentially with the number of features. For example, a binary cut on all features for $d$ features implies $2^d$ outputs per sample. That is why we choose not to run Douglas on datasets with more than 20 features. While the original implementation of Douglas by Yang et al. (2018) is made with Pytorch to benefit from automatic differentiation, we report the result of our own pure-numpy version with explicit derivatives in App. G.

First of all, we observed in Table 3 that Kauri often performs on par with related works. Notably, these performances are close to the KMeans+Tree baseline, except for the digits and wine datasets. Second of all, the performances of related works seem often close to Kauri or slightly below despite

Table 3: ARI scores $_{std}$ (greater is better) of Kauri, Douglas and other methods after 30 runs on random subsamples of 80% of the input datasets. Entries marked X were not run due to memory overflows for Douglas because of the large number of features. All models are limited to finding as many leaves as clusters.

| Dataset | Kauri | KMeans+Tree | Douglas | ExShallow | RDM | IMM |
|---|---|---|---|---|---|---|
| Avila | $0.02_{0.01}$ | $0.04_{0.01}$ | $0.02_{0.01}$ | $\mathbf{0.06_{0.02}}$ | $0.05_{0.02}$ | $\mathbf{0.06_{0.01}}$ |
| Cancer | $0.74_{0.02}$ | $0.73_{0.01}$ | $\mathbf{0.84_{0.02}}$ | $0.74_{0.01}$ | $0.68_{0.02}$ | $0.73_{0.02}$ |
| Car | $0.06_{0.06}$ | $0.08_{0.07}$ | X | $0.05_{0.05}$ | $0.07_{0.05}$ | $0.05_{0.05}$ |
| Congress | $0.49_{0.03}$ | $0.46_{0.04}$ | $\mathbf{0.56_{0.04}}$ | $0.49_{0.03}$ | $0.39_{0.02}$ | $0.48_{0.03}$ |
| Digits | $0.26_{0.02}$ | $\mathbf{0.36_{0.05}}$ | X | $0.31_{0.03}$ | $0.16_{0.03}$ | $0.27_{0.03}$ |
| Haberman | $0.00_{0.03}$ | $0.00_{0.00}$ | $0.02_{0.04}$ | $0.00_{0.00}$ | $0.00_{0.02}$ | $0.00_{0.00}$ |
| Iris | $\mathbf{0.63_{0.07}}$ | $0.60_{0.06}$ | $0.47_{0.12}$ | $0.62_{0.06}$ | $0.49_{0.04}$ | $0.59_{0.05}$ |
| Mice | $\mathbf{0.21_{0.03}}$ | $0.18_{0.04}$ | X | $0.19_{0.03}$ | $0.12_{0.04}$ | $0.16_{0.03}$ |
| Vowel | $0.01_{0.01}$ | $0.03_{0.03}$ | $\mathbf{0.07_{0.05}}$ | $0.05_{0.04}$ | $\mathbf{0.07_{0.03}}$ | $\mathbf{0.08_{0.04}}$ |
| Wine | $0.60_{0.10}$ | $0.71_{0.05}$ | $0.54_{0.13}$ | $\mathbf{0.74_{0.04}}$ | $0.33_{0.05}$ | $\mathbf{0.75_{0.04}}$ |

Table 4: KMeans score $_{std}$ (lower is better) of Kauri and related works after 30 runs on subsamples of 80% of the input datasets divided by the KMeans reference score (=1.0). All models are limited to finding as many leaves as clusters.

| Dataset | Kauri | KMeans+Tree | Douglas | ExShallow | RDM | IMM |
|---|---|---|---|---|---|---|
| Avila | $1.22_{0.08}$ | $1.95_{0.07}$ | $1.72_{0.14}$ | $1.23_{0.10}$ | $1.30_{0.13}$ | $\mathbf{1.15_{0.07}}$ |
| Cancer | $1.08_{0.02}$ | $1.08_{0.02}$ | $\mathbf{1.00_{0.01}}$ | $1.07_{0.02}$ | $1.31_{0.02}$ | $1.07_{0.01}$ |
| Car | $\mathbf{1.00_{0.00}}$ | $\mathbf{1.00_{0.00}}$ | X | $\mathbf{1.00_{0.00}}$ | $1.02_{0.03}$ | $\mathbf{1.00_{0.00}}$ |
| Congress | $1.05_{0.01}$ | $1.04_{0.01}$ | $\mathbf{1.00_{0.01}}$ | $1.04_{0.01}$ | $1.13_{0.02}$ | $1.04_{0.01}$ |
| Digits | $\mathbf{1.13_{0.01}}$ | $1.19_{0.02}$ | X | $\mathbf{1.13_{0.02}}$ | $1.24_{0.04}$ | $1.14_{0.02}$ |
| Haberman | $\mathbf{1.01_{0.00}}$ | $\mathbf{1.01_{0.00}}$ | $1.04_{0.02}$ | $\mathbf{1.01_{0.00}}$ | $\mathbf{1.01_{0.00}}$ | $\mathbf{1.01_{0.00}}$ |
| Iris | $\mathbf{1.06_{0.04}}$ | $1.07_{0.04}$ | $1.49_{0.24}$ | $\mathbf{1.06_{0.05}}$ | $1.29_{0.08}$ | $1.07_{0.05}$ |
| Mice | $\mathbf{1.05_{0.01}}$ | $1.09_{0.03}$ | X | $\mathbf{1.05_{0.01}}$ | $1.33_{0.05}$ | $1.11_{0.03}$ |
| Poker | $\mathbf{1.03_{0.00}}$ | $1.07_{0.02}$ | $1.16_{0.02}$ | $1.05_{0.00}$ | $1.07_{0.02}$ | $1.12_{0.05}$ |
| Vowel | $1.06_{0.00}$ | $1.07_{0.01}$ | $\mathbf{1.04_{0.01}}$ | $1.07_{0.01}$ | $1.09_{0.01}$ | $1.09_{0.01}$ |
| Wine | $1.09_{0.05}$ | $1.13_{0.05}$ | $1.11_{0.09}$ | $\mathbf{1.05_{0.02}}$ | $1.33_{0.05}$ | $\mathbf{1.05_{0.03}}$ |

similar limits in the number of leaves. We believe that this difference can be explained by the order of the choice of splits in the trees owing to the presence of the KMeans objective among methods or just the usage of labels. Regarding the KMeans score in Table 4, Kauri and Douglas both obtain good performances, with scores always at most one standard deviation away from the best model for Kauri.To conclude, we observed encouraging performances from the Douglas algorithm which benefits from the multiple binnings at root level of all features.

## 5.2 ON THE EXPLANATION QUALITY

We are now interested in the explainable nature of the obtained trees. Indeed, several tree structures could easily yield the same clustering and consequently, we need to focus on the explanation quality of the structure.

We provide an example from Moshkovitz et al. (2020) of a non-optimal choice of splits for the KMeans+Tree compared to the optimal found by Kauri in App. F.

We choose to measure the weighted average depth (WAD, Laber et al., 2023) which measures the ratio of samples per leaf multiplied by their respective depth. The lower the WAD, the better the structure of the tree as it yields simpler explanations by being shallow. The benefit of this metric is that it encourages trees to be shallow, a property we seek in the context of limited leaves. This metric cannot be applied however to Douglas because its differentiable tree sets all rules at the same level, i.e. without any notion of path for ordering the rules and leaves. Additionally, we remove for

Table 5: WAD scores $_{std}$ (lower is better) of Kauri and related works after 30 runs on random subsamples of 80% of the input datasets. All models are limited to finding as many leaves as clusters.

| Dataset | Kauri | KMeans+Tree | ExShallow | RDM | IMM |
|---------|-------|-------------|-----------|-----|-----|
| Avila   | $5.47_{0.30}$ | $\mathbf{4.00_{0.13}}$ | $6.43_{0.56}$ | $7.81_{0.33}$ | $9.19_{0.10}$ |
| Car     | $\mathbf{2.00_{0.00}}$ | $2.04_{0.06}$ | $2.05_{0.06}$ | $2.03_{0.08}$ | $2.04_{0.06}$ |
| Digits  | $\mathbf{3.45_{0.22}}$ | $3.48_{0.17}$ | $3.98_{0.19}$ | $5.21_{0.83}$ | $6.79_{0.34}$ |
| Iris    | $1.67_{0.02}$ | $1.67_{0.02}$ | $1.67_{0.02}$ | $\mathbf{1.62_{0.03}}$ | $1.67_{0.02}$ |
| Mice    | $\mathbf{3.04_{0.07}}$ | $3.16_{0.13}$ | $3.23_{0.16}$ | $3.47_{0.39}$ | $4.85_{0.41}$ |
| Poker   | $\mathbf{3.26_{0.00}}$ | $3.26_{0.01}$ | $3.38_{0.05}$ | $\mathbf{3.28_{0.11}}$ | $4.40_{0.45}$ |
| Wine    | $\mathbf{1.58_{0.07}}$ | $1.65_{0.04}$ | $1.69_{0.03}$ | $1.75_{0.03}$ | $1.71_{0.02}$ |

this experiment datasets with 2 clusters as the only way to learn trees on these datasets is to have two leaves at the same depth, yielding a WAD of 1 for all methods.

We give the WAD scores of the previously described benchmark in Table. 5. We observe that Kauri often outperforms related works, even though the KMeans+Tree baseline remains a tough competitor. Moreover, these gains in shallow structure still maintain good clustering on par with related works as seen in the previous section.

To highlight some differences in behaviour between Kauri and KMeans+Tree, we investigate with Figure 2 how the angles of the decision boundary and the number of samples in the dataset can change the performances on seemingly identical distributions. Indeed, KMeans easily builds linear boundaries that are not axis-aligned, hence as the boundaries become less and less aligned with the axes, the decision trees struggle to maintain a low number of leaves to mimic these "diagonal" boundaries. This effect gets worse if the number of samples to separate is high on this decision boundary. However, as soon as the decision boundaries are axis-aligned, the decision tree becomes again a fierce competitor. Both trees have unlimited leaves and only stop when no gain is any longer possible. We use the weighted average explanation size (WAES, Laber et al., 2023) which measures the number of non-redundant rules that define a leaf divided by the number of training samples. The lower the WAES, the better the structure of the tree as it yields simpler explanations. The benefit of this metric is that the depth of a tree matters little, but rather the number of leaves that explain a cluster.

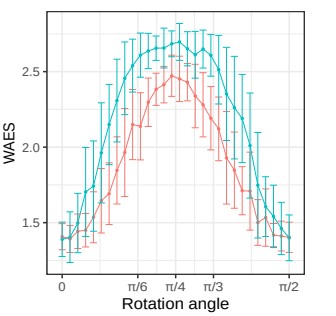 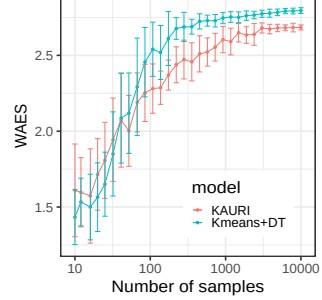 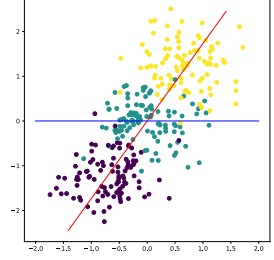

(a) Variations for growing angle, number of samples fixed to 300.

(b) Variations for growing number of samples, angle fixed to $\pi/4$.

(c) Example of dataset for 300 samples and an angle of $\pi/3$

Figure 2: Variations of WAES scores for aligned isotropic 2d Gaussian distributions separated by Kauri or KMeans+Tree as the angle of the alignment (red line in 2c) with the x-axis (blue line in 2c) grows or the number of samples increases over 30 runs. The distance between the means is $\sqrt{2}$ and the scale matrices are $0.2\mathbf{I}_2$.

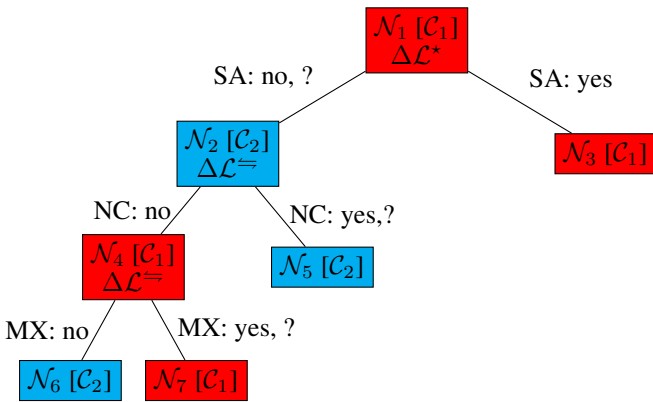

Figure 3: The unsupervised Kauri tree for 2 clusters on the Congressional votes dataset. SA stands for the El Salvador Aid vote, NC for the Nicaraguan Contras vote and MX for the MX-missile vote. The question mark means that the voter did not vote or was missing. Nodes contain their name, the associated cluster to which they assign samples and the type of split that occured during learning.

### 5.3 A QUALITATIVE EXAMPLE OF THE OBTAINED DECISION TREE

In this example, we focus on the congressional votes dataset which details 16 key votes from the 435 members of the US Congress in 1985. The targets of the dataset are the Republican or Democrat affiliations of the voters. We preprocessed the dataset by binarising the vote outcome with $-1$ for "no" and $1$ for "yes". Missing values due to the absence of votes were converted to $0$ which is midway between yes and no and hence does not influence the linear kernel by favouring one type of answer. The Kauri tree that was fitted on this dataset is described in Figure 3. The obtained clusters translate very well the Republican and Democrat opposition through arming and international assistance, with one cluster containing up to 73% of Republicans and the second one adding up to 96% of Democrats. The ARI is 0.47 for this tree which corresponds to an unsupervised accuracy of 84%.

Upon running 30 times the Douglas tree on this dataset, we measure the number of active cut points. The most selected active cut points were on the exact same features as the ones selected by Kauri in Figure 3: the aid to Nicaraguan Contras (selected 93% of time), the El Salvador aid (83%) and the MX missile votes (63%). The models had an average ARI of 0.53.

## 6 FINAL WORDS

We introduced a framework for unsupervised tree end-to-end learning. By combining tree structures with the generalised mutual information for clustering, we derived two novel examples: Kauri and Douglas. The former maximises a kernel-KMeans-like objective to build iteratively unsupervised splits through the affectation of tree leaves to existing or new clusters while the latter exploits the combined potential of differential trees and the Wasserstein distance. Kauri can be privileged for small-scale datasets whereas Douglas is better suited for long datasets on condition of few features. Overall, both methods achieve good performances in clustering with Kauri being on par with related works for unsupervised trees using shallower trees. The strong advantage of these methods is building an interpretable by-nature clustering instead of seeking to explain another clustering output from a different algorithm. Finally, we think that the combination of KMeans and a decision tree remains a strong baseline that should be provided in works on unsupervised tree works.

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

## A   SIMILARITIES WITH HIERARCHICAL CLUSTERING

Trees are closely related to divisive hierarchical clustering algorithms. Indeed, divisive algorithms are built top-down (Roux, 2015) and if we give up the interpretability of trees by not restraining the splits on data-dependent thresholds, we can sort the data per node anyhow for the best split. In such case, the proposals of split become combinatorial and reach $2^{n-1} - 1$ proposals for $n$ samples (Edwards & Cavalli-Sforza, 1965). These models are therefore costly, yet account for few iterations as we only need $K - 1$ splits to obtain $K$ clusters.

Multiple splitting procedures were proposed to alleviate computation of all splits (Williams & Lambert, 1959; Hubert, 1973) to even KMeans (Mollineda & Vidal, 2000). One of the most known divisive clustering algorithms is DIANA (Kaufman & Rousseeuw, 1990). Roux (2015) suggests that ratio-based splits are among the most efficient (Roux, 2015).

To further accelerate the evaluation of the gains procedures and constraining the search space, combinations of divisive clustering algorithms with model-based clustering algorithms were proposed, hence using the maximum likelihood as a global objective for the model (Sharma et al., 2017; Burghardt et al., 2022). We can still note the usage of KMeans as a heuristic for proposing splits at each node (Sharma et al., 2017). However, the improvements brought by model-based clustering require parametric hypotheses which may constrain the exploration of the tree.

## B   DERIVATIONS OF THE KAURI OBJECTIVE AND ITS RELATIONSHIP TO KMEANS

We show here how to derive the Kauri objective function from both the one-vs-all and one-vs-one squared-MMD-GEMINI before showing its relationship to the minimisation of a kernel KMeans objetive.

### B.1   DERIVING AN OBJECTIVE FROM THE MMD-GEMINI

#### B.1.1   MMD-OVA

We start by recovering the definition of the OvA squared-MMD-GEMINI using only the outputs of the outputs of our model $p_\theta(y|\boldsymbol{x})$:

$$\mathcal{I}_{\mathrm{MMD}^2}^{\mathrm{ova}}(\boldsymbol{x}; y|\theta) = \mathbb{E}_{y \sim p_\theta(y)} \left\{ \mathbb{E}_{\boldsymbol{x}_a, \boldsymbol{x}_b \sim p(\boldsymbol{x})} \left[ \kappa(\boldsymbol{x}_a, \boldsymbol{x}_b) \left( \frac{p_\theta(y|\boldsymbol{x}_a)p_\theta(y|\boldsymbol{x}_b)}{p_\theta(y)^2} + 1 - 2\frac{p_\theta(y|\boldsymbol{x}_a)}{p_\theta(y)} \right) \right] \right\}. \tag{11}$$

We can replace the expectations with discrete sums for both the clusters and the data. Notice the factor $\frac{1}{n}$ in front of the kernel as we are simply doing a Monte Carlo estimate. We replace at the same time, the values of the distributions by either the indicator functions or cluster sizes:

$$\mathcal{I}_{\mathrm{MMD}^2}^{\mathrm{ova}}(\boldsymbol{x}; y|\theta) = \sum_{k=1}^{K} \frac{|\mathcal{C}_k|}{n} \sum_{\substack{i=1 \\ j=1}}^{n} \frac{\kappa(\boldsymbol{x}_i, \boldsymbol{x}_j)}{n^2} \left( \frac{\mathbb{1}[\boldsymbol{x}_i \in \mathcal{X}_k]\mathbb{1}[\boldsymbol{x}_j \in \mathcal{X}_k]n^2}{|\mathcal{C}_k|^2} + 1 - 2\frac{\mathbb{1}[\boldsymbol{x}_i \in \mathcal{X}_k]n}{|\mathcal{C}_k|} \right). \tag{12}$$

By applying the indicator functions, we see that we sum the terms of a kernel on condition that the respective samples belong specifically to some subset of data. We can consequently rewrite the inner sum as a combination of *kernel stocks* $\sigma$:

$$\mathcal{I}_{\mathrm{MMD}^2}^{\mathrm{ova}}(\boldsymbol{x}; y|\theta) = \sum_{k=1}^{K} \frac{|\mathcal{C}_k|}{n} \left( \frac{\sigma(\mathcal{C}_k^2)}{|\mathcal{C}_k|^2} + \frac{\sigma([n]^2)}{n^2} - 2\frac{\sigma(\mathcal{C}_k \times [n])}{n|\mathcal{C}_k|} \right). \tag{13}$$

Here, $\sigma([n]^2) = \sigma(\{1, \cdots, n\} \times \{1, \cdots, n\})$ is simply the *kernel stock* of the complete dataset, i.e. the sum of all kernel elements of the dataset. Then, a couple of simplifications give:

$$\mathcal{I}_{\mathrm{MMD}^2}^{\mathrm{ova}}(\boldsymbol{x}; y|\theta) = \sum_{k=1}^{K} \frac{\sigma(\mathcal{C}_k^2)}{n|\mathcal{C}_k|} + \frac{|\mathcal{C}_k|\sigma([n]^2)}{n^3} - 2\frac{\sigma(\mathcal{C}_k \times [n])}{n^2}. \tag{14}$$

We can obtain the final form of the GEMINI by summing constant terms. Observing that $\sum_k |\mathcal{C}_k| = n$ and using the bilinearity of $\sigma$, we have:

$$\mathcal{I}^{\text{ova}}_{\text{MMD}^2}(\boldsymbol{x}; y|\theta) = -\frac{\sigma([n]^2)}{n^2} + \sum_{k=1}^{K} \frac{\sigma(\mathcal{C}_k^2)}{n|\mathcal{C}_k|}. \tag{15}$$

As we are interested in optimising the clustering assignments in the tree, we can remove all constant terms and factors that do not bring extra information. Hence, the final objective $\mathcal{L}$ to maximise is:

$$\mathcal{L} = \sum_{k=1}^{K} |\mathcal{C}_k|^{-1} \sigma(\mathcal{C}_k^2) \tag{16}$$

### B.1.2 MMD-OvO

We now prove that the objective function $\mathcal{L}$ obtained in the previous section is also equivalent to maximising the OvO squared MMD-GEMINI in case of delta Dirac classifiers. We start by expressing the complete squared MMD GEMINI:

$$\mathcal{I}^{\text{ovo}}_{\text{MMD}^2}(\boldsymbol{x}; y|\theta) = \mathbb{E}_{y_a, y_b \sim p_\theta(y)} \left[ \mathbb{E}_{\boldsymbol{x}_a, \boldsymbol{x}_b \sim p(\boldsymbol{x})} \left[ \kappa(\boldsymbol{x}_a, \boldsymbol{x}_b) \left( \frac{p_\theta(y_a|\boldsymbol{x}_a)p_\theta(y_a|\boldsymbol{x}_b)}{p_\theta(y_a)^2} \right. \right. \right.$$
$$\left. \left. \left. + \frac{p_\theta(y_b|\boldsymbol{x}_a)p_\theta(y_b|\boldsymbol{x}_b)}{p_\theta(y_b)^2} - 2\frac{p_\theta(y_a|\boldsymbol{x}_a)p_\theta(y_b|\boldsymbol{x}_b)}{p_\theta(y_a)p_\theta(y_b)} \right) \right] \right]. \tag{17}$$

As we exactly did for the OvA MMD-GEMINI demonstration, we apply the following tricks: discretising the expectations on the dataset $\mathcal{D}$, re-expressing the cluster proportions, simplifying the sums. Our discrete version is:

$$\mathcal{I}^{\text{ovo}}_{\text{MMD}^2}(\boldsymbol{x}; y|\theta) = \sum_{k,k'}^{K} \frac{|\mathcal{C}_k||\mathcal{C}_{k'}|}{n^2} \sum_{\substack{x_i \in \mathcal{D} \\ x_j \in \mathcal{D}}} \frac{\kappa(\boldsymbol{x}_i, \boldsymbol{x}_j)}{n^2} \left( \frac{n^2 \mathbb{1}[\boldsymbol{x}_i \in \mathcal{X}_k]\mathbb{1}[\boldsymbol{x}_j \in \mathcal{X}_k]}{|\mathcal{C}_k|^2} \right.$$
$$\left. + \frac{n^2 \mathbb{1}[\boldsymbol{x}_i \in \mathcal{X}_{k'}]\mathbb{1}[\boldsymbol{x}_j \in \mathcal{X}_{k'}]}{|\mathcal{C}_{k'}|^2} - 2\frac{n^2 \mathbb{1}[\boldsymbol{x}_i \in \mathcal{X}_k]\mathbb{1}[\boldsymbol{x}_j \in \mathcal{X}_{k'}]}{|\mathcal{C}_k||\mathcal{C}_{k'}|} \right). \tag{18}$$

We can cancel the factors in $n^2$ and replace our sum over indicator functions by the *kernel stock* function $\sigma$. Thus, we obtain:

$$\mathcal{I}^{\text{ovo}}_{\text{MMD}^2}(\boldsymbol{x}; y|\theta) = \sum_{k,k'}^{K} \frac{|\mathcal{C}_k||\mathcal{C}_{k'}|}{n^2} \left( \frac{\sigma(\mathcal{C}_k \times \mathcal{C}_k)}{|\mathcal{C}_k|^2} + \frac{\sigma(\mathcal{C}_{k'} \times \mathcal{C}_{k'})}{|\mathcal{C}_{k'}|^2} - 2\frac{\sigma(\mathcal{C}_k \times \mathcal{C}_{k'})}{|\mathcal{C}_k||\mathcal{C}_{k'}|} \right). \tag{19}$$

For the first two terms, we can cancel one part of the summation. Indeed, the *kernel stock* on $\mathcal{C}_k$ does not depend on $k'$, consequently, the sum over $k'$ just multiplies this *kernel stock* up to a factor $n$ which will be cancelled by the denominator $n$ at the very start. The same reasoning goes for the second term. Last but not least, we can cancel cluster sizes on the last term. Our expression is then:

$$\mathcal{I}^{\text{ovo}}_{\text{MMD}^2}(\boldsymbol{x}; y|\theta) = 2\sum_{k=1}^{K} \frac{\sigma(\mathcal{C}_k \times \mathcal{C}_k)}{n|\mathcal{C}_k|} - 2\sum_{k,k'}^{K} \frac{\sigma(\mathcal{C}_k \times \mathcal{C}_{k'})}{n^2}. \tag{20}$$

As we now look forward to maximising this expression, we can realise that the last term is simply the full *kernel stock*, in other words, a constant with respect to the clustering. We can then discard this term. For the first term, we simply remove the constant factors 2 and $1/n$ to obtain the equivalent:

$$\mathcal{I}_{\text{MMD}^2}^{\text{ovo}}(\boldsymbol{x}; y|\theta) = \frac{2}{n}\left(\sum_{k=1}^{K}\frac{\sigma(\mathcal{C}_k \times \mathcal{C}_k)}{|\mathcal{C}_k|} - \frac{\sigma(\mathcal{D} \times \mathcal{D})}{n}\right) \propto \mathcal{L} + \text{constant.} \tag{21}$$

This concludes our proof.

### B.2 RELATIONSHIP TO KERNEL KMEANS

An alternative formulation of the kernel KMeans objective is:

$$\theta^\star = \arg\min_\theta \sum_{k=1}^{K}\frac{1}{|\mathcal{C}_k|}\sum_{\boldsymbol{x},\boldsymbol{y}\in\mathcal{C}_k}\|\varphi(\boldsymbol{x}) - \varphi(\boldsymbol{y})\|_{\mathcal{H}}^2, \tag{22}$$

where $\mathcal{H}$ is a Hilbert space and $\varphi$ its projection. By rephrasing this objective in terms of kernel, we can obtain the following equation:

$$\theta^\star = \arg\min_\theta \sum_{k=1}^{K}\frac{1}{|\mathcal{C}_k|}\sum_{\boldsymbol{x},\boldsymbol{y}\in\mathcal{C}_k}\kappa(\boldsymbol{x},\boldsymbol{x}) + \kappa(\boldsymbol{y},\boldsymbol{y}) - \kappa(\boldsymbol{x},\boldsymbol{y}), \tag{23}$$

where the two first kernel terms can be summarised as the size of clusters weighting the diagonal elements of the kernel. Finally, the third term is the *kernel stock* $\sigma$ of the cluster, and thus:

$$\theta^\star = \arg\min_\theta \sum_{\boldsymbol{x}\in\mathcal{D}}\kappa(\boldsymbol{x},\boldsymbol{x}) - \sum_{k=1}^{K}\frac{1}{|\mathcal{C}_k|}\sigma(\mathcal{C}_k \times \mathcal{C}_k). \tag{24}$$

Therefore, maximising our objective function $\mathcal{L}$ is equivalent up to a constant to minimising a KMeans objective for any kernel. This joins the observation of França et al. (2020) who connected an objective similar to a one-vs-one squared MMD to the kernel KMeans. Notice that we removed a factor 2 in the final equation as it does not affect the argmin operator.

## C DEMONSTRATION OF EXPRESSIONS FOR THE KAURI GAINS

We can derive from our objective (Eq. 6) four metrics which correspond to different types of splits: the *star gain*, the *double star gain*, the *switch gain* and the *reallocation gain*. We provide Figure 4 for visual purpose and assistance in the demonstrations.

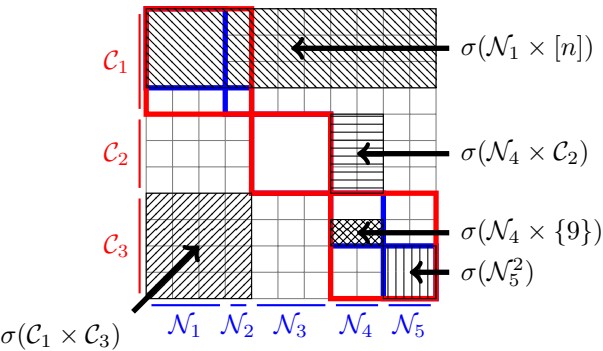

Figure 4: A toy example with a dataset consisting of 11 samples partitioned in 3 clusters using 5 leaves in a tree. The matrix represents the kernel between all pairs of samples and dashed areas correspond to the sum of kernel elements according to the *kernel stock* function $\sigma$.

### C.1 CREATING A NEW CLUSTER: THE *star gain*

In this case, we assign one of the splits $\mathcal{S}_L$ or $\mathcal{S}_R$ to a new cluster and let the other split in the same cluster as the parent node, i.e. either $k_L = K+1$ and $k_R = k_p$ or $k_L = k_p$ and $k_R = K+1$. Taking the case where the left split is given to a new cluster, we derive from the global gain a variation that we call *star gain*:

$$\Delta\mathcal{L}^{\star}(\mathcal{S}_L : k_p \to k_L) = \frac{\sigma(\mathcal{S}_L^2)}{|\mathcal{S}_L|} + \frac{\sigma(\mathcal{C'}_{k_p}^{\ 2})}{|\mathcal{C'}_{k_p}|} - \frac{\sigma(\mathcal{C}_{k_p}^2)}{|\mathcal{C}_{k_p}|}. \tag{25}$$

However, that expression is not convenient since there is a clear dependence: $\mathcal{C'}_{k_p} = \mathcal{C}_{k_p} \setminus \mathcal{S}_L$ and we would be interested in avoiding the evaluation of $\sigma(\mathcal{C'}_{k_p}^{\ 2})$. We can use the bilinearity of the sigma function and decompose over the new cluster $\mathcal{C'}_{k_p} = \mathcal{C}_{k_p} \setminus \mathcal{S}_L$. Similarly, we can reexpress the cardinal as $|\mathcal{C'}_{k_p}| = |\mathcal{C}_{k_p}| - |\mathcal{S}_L|$. Consequently, our term becomes:

$$\Delta\mathcal{L}^{\star}(\mathcal{S}_L : k_p \to k_L) = \frac{\sigma(\mathcal{S}_L^2)}{|\mathcal{S}_L|} + \frac{\sigma(\mathcal{C}_{k_p}^2) - 2\sigma(\mathcal{C}_{k_p} \times \mathcal{S}_L) + \sigma(\mathcal{S}_L^2)}{|\mathcal{C}_{k_p}| - |\mathcal{S}_L|} - \frac{\sigma(\mathcal{C}_{k_p}^2)}{|\mathcal{C}_{k_p}|}. \tag{26}$$

It is then just a matter of reordering with respect to the *kernel stocks* $\sigma$ to obtain the final equation:

$$\Delta\mathcal{L}^{\star}(\mathcal{S}_L : k_p \to k_L) = \sigma(\mathcal{S}_L^2)\left(\frac{1}{|\mathcal{S}_L|} + \frac{1}{|\mathcal{C}_{k_p}| - |\mathcal{S}_L|}\right) + \sigma(\mathcal{C}_{k_p}^2)\left(\frac{1}{|\mathcal{C}_{k_p}| - |\mathcal{S}_L|} - \frac{1}{|\mathcal{C}_{k_p}|}\right)$$
$$- 2\frac{\sigma(\mathcal{C}_{k_p} \times \mathcal{S}_L)}{|\mathcal{C}_{k_p}| - |\mathcal{S}_L|}, \tag{27}$$

which will be the used equation for the *star gain*.

### C.2 CREATING TWO CLUSTERS: THE *double star gain*

The operation of creating two clusters can be seen as assigning in a first step the complete node $p$ to a new cluster, then taking one of its split and assigning it to a second new cluster. The *double star gain* $\Delta\mathcal{L}^{\star\star}$ can be thus computed by the sum of $\Delta\mathcal{L}^{\star}$ with $\mathcal{N}_p$ replacing the source cluster $\mathcal{C}_{k_p}$ and another $\Delta\mathcal{L}^{\star}$ with the $\mathcal{N}_p$ replacing $\mathcal{S}_L$:

$$\Delta\mathcal{L}^{\star\star}(\mathcal{S}_L \to k_L, \mathcal{S}_R \to k_R) = \Delta\mathcal{L}^{\star}(\mathcal{N}_p : k_p \to k_L) + \Delta\mathcal{L}^{\star}(\mathcal{S}_R : k_L \to k_R) \tag{28}$$

#### C.2.1 MERGING WITH ANOTHER CLUSTER: THE *switch gain*

This type of split is very similar to the creation of a new one. The main difference is that as one of the child nodes will join another cluster, e.g. $k_p \neq k_L \leq K$, we must take into account in the gain that we must subtract the *kernel stock* of the former target cluster. We call this type of gain the *switch gain*:

$$\Delta\mathcal{L}^{\rightleftarrows}(\mathcal{S}_L : k_p \to k_L) = \frac{\sigma(\mathcal{C'}_{k_L}^{\ 2})}{|\mathcal{C'}_{k_L}|} + \frac{\sigma(\mathcal{C'}_{k_p}^{\ 2})}{|\mathcal{C'}_{k_p}|} - \frac{\sigma(\mathcal{C}_{k_L}^2)}{|\mathcal{C}_{k_L}|} - \frac{\sigma(\mathcal{C}_{k_p}^2)}{|\mathcal{C}_{k_p}|}, \tag{29}$$

where we arbitrarily chose the left split for the equation. Similarly to the new cluster case, this expression can be completely re-written using only the original clusters and $\mathcal{S}_L$ to remove dependencies in the equation. We start by exploiting the bilinearity of $\sigma$. The first new cluster is the source one without the split elements and the second new cluster is the target one with added split elements. Therefore, we have $\mathcal{C'}_{k_p} = \mathcal{C}_{k_p} \setminus \mathcal{S}_L$ and $\mathcal{C'}_{k_L} = \mathcal{C}_{k_L} \cup \mathcal{S}_L$. We can deduce:

$$\Delta\mathcal{L}^{\rightleftarrows}(\mathcal{S}_L : k_p \to k_L) = \frac{\sigma(\mathcal{C}_{k_p}^2) - 2\sigma(\mathcal{C}_{k_p} \times \mathcal{S}_L) + \sigma(\mathcal{S}_L^2)}{|\mathcal{C}_{k_p}| - |\mathcal{S}_L|} + \frac{\sigma(\mathcal{C}_{k_L}^2) + 2\sigma(\mathcal{C}_{k_L} \times \mathcal{S}_L) + \sigma(\mathcal{S}_L^2)}{|\mathcal{C}_{k_L}| + |\mathcal{S}_L|}$$
$$- \frac{\sigma(\mathcal{C}_{k_p}^2)}{|\mathcal{C}_{k_p}|} - \frac{\sigma(\mathcal{C}_{k_L}^2)}{|\mathcal{C}_{k_L}|}. \tag{30}$$

Then we finish again the demonstration by reordering the factors according to the respective stocks:

$$\Delta\mathcal{L}^{\rightleftarrows}(\mathcal{S}_L : k_p \rightarrow k_L) = \sigma(\mathcal{S}_L^2)\left(\frac{1}{|\mathcal{C}_{k_L}| + |\mathcal{S}_L|} + \frac{1}{|\mathcal{C}_{k_p}| - |\mathcal{S}_L|}\right) - 2\frac{\sigma(\mathcal{C}_{k_p} \times \mathcal{S}_L)}{|\mathcal{C}_{k_p}| - |\mathcal{S}_L|}$$

$$+ \sigma(\mathcal{C}_{k_p}^2)\left(\frac{1}{|\mathcal{C}_{k_p}| - |\mathcal{S}_L|} - \frac{1}{|\mathcal{C}_{k_p}|}\right) + \sigma(\mathcal{C}_{k_L}^2)\left(\frac{1}{|\mathcal{C}_{k_L}| + |\mathcal{S}_L|} - \frac{1}{|\mathcal{C}_{k_p}|}\right) + 2\frac{\sigma(\mathcal{C}_{k_L} \times \mathcal{S}_L)}{|\mathcal{C}_{k_L}| + |\mathcal{S}_L|}. \quad (31)$$

which is the equation we use in Kauri switch splits.

### C.3 Reallocating content to the different clusters: the *reallocation gain*

As the tree grows, it may as well be interesting to reconsider whether samples that are currently in a cluster should all be in a new cluster. We call this process *reallocation* as both splits of a node end up in two different clusters: $k_L \neq k_p$ and $k_R \neq k_p$.

Contrary to the double cluster creation case, we cannot simply sum two *switch gains* $\Delta\mathcal{L}^{\rightleftarrows}$ to compute the *reallocation gain*. Indeed, the *switch gain* assumes that the final state of the original cluster $\mathcal{C}_{k_p}$ still contains the complementary of the chosen split $\mathcal{S}_L$ (or $\mathcal{S}_R$) from the leaf samples $\mathcal{N}_p$ which is not true when both parts of the leaf go to different clusters. Hence, a corrective term $\epsilon$ is required.

When we sum two switch gains, the final state of the target clusters is correct: we simply added elements from a split. The corrective term thus only focuses on the state of the source cluster. Let $\mathcal{C}'_k$ the state of the source cluster according to the first switch gain on the left split, $\mathcal{C}''_k$ the state of the source cluster according to the second switch gain on the right split and $\mathcal{C}_k^{\hookrightarrow}$ the true state after reallocating both left and right splits. Notice that we lighten the notation $k_p$ to $k$. The corrective term must satisfy:

$$\frac{\sigma(\mathcal{C}_k'^2)}{|\mathcal{C}'_k|} + \frac{\sigma(\mathcal{C}_k''^2)}{|\mathcal{C}''_k|} + \epsilon = \frac{\sigma(\mathcal{C}_k^{\hookrightarrow 2})}{|\mathcal{C}_k^{\hookrightarrow}|}. \quad (32)$$

We can rewrite each new definition of the source clusters using the left split $\mathcal{S}_L$ and right split $\mathcal{S}_R$. Thus we get:

$$\frac{\sigma(\mathcal{C}_k \setminus \mathcal{S}_L^2)}{|\mathcal{C}_k| - |\mathcal{S}_L|} + \frac{\sigma(\mathcal{C}_k \setminus \mathcal{S}_R^2)}{|\mathcal{C}_k| - |\mathcal{S}_R|} + \epsilon = \frac{\sigma(\mathcal{C}_k \setminus \mathcal{N}_p^2)}{|\mathcal{C}_k| - |\mathcal{N}_p|}, \quad (33)$$

which allows us to use the bilinearity of $\sigma$:

$$\frac{\sigma(\mathcal{C}_k^2) - 2\sigma(\mathcal{C}_k \times \mathcal{S}_L) + \sigma(\mathcal{S}_L^2)}{|\mathcal{C}_k| - |\mathcal{S}_L|} + \frac{\sigma(\mathcal{C}_k^2) - 2\sigma(\mathcal{C}_k \times \mathcal{S}_R) + \sigma(\mathcal{S}_R^2)}{|\mathcal{C}_k| - |\mathcal{S}_R|} + \epsilon$$

$$= \frac{\sigma(\mathcal{C}_k^2) - 2\sigma(\mathcal{C}_k \times \mathcal{N}_p) + \sigma(\mathcal{N}_p^2)}{|\mathcal{C}_k| - |\mathcal{N}_p|}. \quad (34)$$

Notice that we use the simplification $\mathcal{N}_p = \mathcal{S}_L \cup \mathcal{S}_R$ since the node samples $\mathcal{N}_p$ are divided into two subsets. Then, by reordering the terms and simplifying for the factor $\sigma(\mathcal{C}_k^2)$, we get the expression of $\epsilon$:

$$\epsilon = \frac{\sigma(\mathcal{C}_{k_p}^2) + \sigma(\mathcal{N}_p^2) - 2\sigma(\mathcal{C}_{k_p} \times \mathcal{N})}{|\mathcal{C}_{k_p}| - |\mathcal{N}_p|} + \frac{\sigma(\mathcal{C}_{k_p}^2)}{|\mathcal{C}_{k_p}|} - \frac{\sigma(\mathcal{C}_{k_p}^2) + \sigma(\mathcal{S}_L^2) - 2\sigma(\mathcal{C}_{k_p} \times \mathcal{S}_L)}{|\mathcal{C}_{k_p}| - |\mathcal{S}_L|}$$

$$- \frac{\sigma(\mathcal{C}_{k_p}^2) + \sigma(\mathcal{S}_R^2) - 2\sigma(\mathcal{C}_{k_p} \times \mathcal{S}_R)}{|\mathcal{C}_{k_p}| - |\mathcal{S}_R|}. \quad (35)$$

Thus, we can express the *reallocation gain* $\Delta\mathcal{L}^{\hookrightarrow}$ as the sum of two switch gains assigning both left and right children nodes to different clusters plus the corrective term $\epsilon$.

$$\Delta\mathcal{L}^{\hookrightarrow}(\mathcal{S}_L : k_p \rightarrow k_L, \mathcal{S}_R : k_p \rightarrow k_R) = \Delta\mathcal{L}^{\rightleftarrows}(\mathcal{S}_L : k_p \rightarrow k_L) + \Delta\mathcal{L}^{\rightleftarrows}(\mathcal{S}_L : k_p \rightarrow k_R) + \epsilon \quad (36)$$

# D  A FAST IMPLEMENTATION FOR KAURI

## D.1  THE KAURI ALGORITHM

Upon looking at one leaf containing a small subset of samples, we need to find the best possible split according to a given threshold on a specified feature. While each feature specifies a different ordering and offers little space for optimisation, computing all possible gains may be time-consuming. Indeed, computing $\sigma(E \times F)$ is done in $\mathcal{O}(|E||F|)$, so evaluating gains for a proposal split $\mathcal{S}$ on a single feature for node samples $\mathcal{N}_p$ contributing to a cluster of $\mathcal{C}_k$ has a naive complexity of: $\mathcal{O}(|\mathcal{S}|^2 + |\mathcal{C}_k|^2 + |\mathcal{C}_k||\mathcal{S}|)$ for $\Delta\mathcal{L}^\star$, $\mathcal{O}(|\mathcal{S}|^2 + |\mathcal{C}_k|^2 + |\mathcal{N}_p|^2 + |\mathcal{C}_k|(|\mathcal{S}| + |\mathcal{N}_p|))$ for $\Delta\mathcal{L}^{\star\star}$, $\mathcal{O}(|\mathcal{N}_p|^2 + |\mathcal{C}_k|^2 + \mathcal{C}_k||\mathcal{N}_p| + |\mathcal{C}_{k'}|^2 + |\mathcal{C}_{k'}||\mathcal{N}_p|)$ for each $k' \neq k$ for $\Delta\mathcal{L}^{\rightleftarrows}$, and at worst $K$ times the previous complexity again for all pairs of assignable new clusters $k'$, $k''$ in the *reallocation gain*. Therefore, iterating over all features and all possible splits needs to be optimised as this operation is the core of the tree construction.

### D.1.1  PRE-COMPUTING KERNEL STOCKS

Most of the kernel stocks can be computed ahead in fact, and then the splitting choice would just need to access the value of the kernel stocks instead. To that end, we choose to formulate two matrices that will store all structural information. The matrix $\boldsymbol{Z} \in \{0,1\}^{L_{\max} \times n}$ describes the membership of samples to leaves where $L_{\max}$ is the maximal number of leaves allowed ($L_{\max} \leq n$). As a sample can only belong to 1 leaf, each column of $\boldsymbol{Z}$ has a single 1. Similarly, the matrix $\boldsymbol{Y} \in \{0,1\}^{K_{\max} \times L_{\max}}$ describes the membership of leaves to clusters, and only one cluster is allowed per leaf. We can then compute most of the kernel stocks required for split computations, as:

$$\boldsymbol{\Lambda} = [\sigma(\mathcal{N}_i \times \{x_j\})] = \boldsymbol{Z}\boldsymbol{\kappa}, \tag{37}$$

is the matrix containing all stocks between leaves and single samples requiring $\mathcal{O}(n^2 L_{\max})$ to compute, and:

$$\boldsymbol{\gamma} = [\sigma(\mathcal{C}_i \times \mathcal{C}_j)] = \boldsymbol{Y}\boldsymbol{\Lambda}\boldsymbol{Z}^\top\boldsymbol{Y}^\top, \tag{38}$$

is the matrix with cluster-cluster stocks, requiring $\mathcal{O}(n^2 L_{\max} + L_{\max}^2 K_{\max})$ for computations.

### D.1.2  OPTIMISING SPLIT EVALUATION

Thanks to the formulation of the *star gain* $\Delta\mathcal{L}^\star$, the *double star gain* $\Delta\mathcal{L}^{\star\star}$ and the storage of dynamic evolution of the variables $\sigma(\mathcal{S}_L \times \mathcal{S}_l)$ (resp. $\mathcal{S}_R$) and $\sigma(\mathcal{S}_L \times \mathcal{C}_k)$ (resp. $\mathcal{S}_R$), evaluating the creation of clusters is done in $\mathcal{O}(1)$. Inevitably, we achieve $\mathcal{O}(K)$ for the switch gains $\Delta\mathcal{L}^{\rightleftarrows}$ since evaluating these gains is easy but needs iteration over all clusters.

To alleviate the complexity of the *reallocation gain* $\Delta\mathcal{L}^{\hookrightarrow}$ due to the exploration of all pairs of clusters, we propose to remember the top two switch gains per left children and right children. Indeed, the corrective term $\epsilon$ does not depend on the two clusters to which the left and right children will be reallocated. Hence maximising the *reallocation gain* is the same as finding the combination of the best switch gains. Thus, remembering the top two switch gains and finding the best combination between left and right child, with different clusters membership per child, will yield the optimal *reallocation gain*. Therefore we achieved the best gain in $\mathcal{O}(K)$ and the evaluation of all types of gain is done in $\mathcal{O}(K)$.

### D.1.3  AN ITERATIVE RULE FOR SPLIT STOCKS

Starting from here, we seek an update rule that allows us to easily update the kernel stocks of the splits $\sigma(\mathcal{S} \times \mathcal{C}_k)$ and $\sigma(\mathcal{S}^2)$. We must explore all possible splits by considering thresholds on chosen variables. Therefore, a split on a variable must be done according to the ordering imposed by that variable, leaving a left child $\mathcal{S}^L$ and a right child $\mathcal{S}^R$. The algorithm should ideally consist in starting from the split of a single sample to the left child $\mathcal{S}_1^L$ and all other samples to the right child $\mathcal{S}_1^R$ and progressively add samples according to an ordering given by a sorted feature: $t = \nu(l)$ to compute

---

**Algorithm 1** Finding the best split according a feature-specified ordering $\nu$ at a given node $\mathcal{N}$

---

**Require:** $\mathcal{N}$ the set of indices of samples in the leaf of length $|\mathcal{N}|$
**Require:** $p$ the index of the leaf
**Require:** $\nu$ an ordering precised by a feature of length $|\mathcal{N}|$
**Require:** $\boldsymbol{\kappa}$ a kernel of shape $n \times n$
**Require:** $\boldsymbol{\Lambda}$ the $L_{\max} \times n$ leaf-sample stocks
**Require:** $\boldsymbol{\omega}$ the $K_{\max} \times n$ cluster-sample stocks
**Require:** $\boldsymbol{\gamma}$ the $K_{\max} \times K_{\max}$ cluster-cluster stocks
**Require:** $|\mathcal{C}_k|, \forall k$ the size of all clusters
**Require:** $k$ the cluster of the considered leaf

1: **function** FINDBESTSPLIT($\mathcal{N}, \nu, \boldsymbol{\kappa}, \Lambda, \boldsymbol{\omega}, \boldsymbol{\gamma}, \{|\mathcal{C}_k|\}, k$)
2: $\quad \sigma(\mathcal{S}_L^0 \times \mathcal{S}_L^0) \leftarrow 0$ $\qquad\qquad\qquad\qquad\qquad$ ▷ Initialise all iteration variables
3: $\quad \sigma(\mathcal{S}_R^0 \times \mathcal{S}_R^0) \leftarrow \sum_{i \in \mathcal{N}} \boldsymbol{\Lambda}_{pi}$
4: $\quad \sigma(\mathcal{S}_L^0 \times \mathcal{C}_k) \leftarrow 0, \forall k \leq K_{\max}$ $\qquad\qquad\qquad$ ▷ Arrays of size $K_{\max}$
5: $\quad \sigma(\mathcal{S}_R^0 \times \mathcal{C}_k) \leftarrow \sum_{i \in \mathcal{N}} \boldsymbol{\omega}_{ki}, \forall k \leq K_{\max}$
6: $\quad \sigma(\mathcal{N} \times \mathcal{N}) \leftarrow \sigma(\mathcal{S}_R^0 \times \mathcal{S}_R^0)$
7: $\quad K \leftarrow |\{k \text{ s.t. } |\mathcal{C}_k| \neq 0\}|$ $\qquad\qquad\qquad$ ▷ Current number of clusters
8: $\quad \Delta\mathcal{L}, \texttt{BestSplit} \leftarrow 0, \emptyset$ $\qquad\qquad\qquad\qquad$ ▷ Best split so far
9: $\quad$ **for** $l \leftarrow 1$ to $|\mathcal{N}| - 1$ **do**
10: $\qquad \alpha, \beta \leftarrow 0,0$
11: $\qquad$ **for** $l' \leftarrow 1$ to $|\mathcal{N}|$ **do**
12: $\qquad\quad$ **if** $l' < l$ **then**
13: $\qquad\qquad \alpha \leftarrow \alpha + \kappa_{\nu(l),\nu(l')}$
14: $\qquad\quad$ **end if**
15: $\qquad\quad$ **if** $l' > l$ **then**
16: $\qquad\qquad \beta \leftarrow \beta + \kappa_{\nu(l),\nu(l')}$
17: $\qquad\quad$ **end if**
18: $\qquad$ **end for**
19: $\qquad$ Update $\sigma(\mathcal{S}_L^l \times \mathcal{S}_L^l), \sigma(\mathcal{S}_R^l \times \mathcal{S}_R^l), \sigma(\mathcal{S}_L^l \times \mathcal{C}_k)$ and $\sigma(\mathcal{S}_R^l \times \mathcal{C}_k)$ using equations 46, 47,
$\qquad$ 48, 49.
20: $\qquad k_L^\star, k_R^\star, \texttt{type}^\star \leftarrow \underset{k_L,k_R,\texttt{type}=\{\star,\star\star,\rightleftarrows,\hookrightarrow\}}{\arg\max} \Delta\mathcal{L}^{\texttt{type}}(\mathcal{S}_L : k \mapsto k_L, \mathcal{S}_R : k \mapsto k_R)$
21: $\qquad \tilde{\Delta\mathcal{L}} \leftarrow \Delta\mathcal{L}^{\texttt{type}^\star}(\mathcal{S}_L : k \mapsto k_L^\star, \mathcal{S}_R : k \mapsto k_R^\star)$
22: $\qquad$ **if** $\tilde{\Delta\mathcal{L}} > \Delta\mathcal{L}$ **then**
23: $\qquad\quad \Delta\mathcal{L} \leftarrow \tilde{\Delta\mathcal{L}}$
24: $\qquad\quad \texttt{BestSplit} \leftarrow (k_L^\star, k_R^\star, \nu(l))$ ▷ The split gives the left target, the right target, the
$\qquad$ sample on which the split is done
25: $\qquad$ **end if**
26: $\quad$ **end for**
27: $\quad$ **return** $\Delta\mathcal{L}, \texttt{BestSplit}$
28: **end function**

---

$\mathcal{S}_l^L$ and $\mathcal{S}_l^R$. For example, if the $p$-th node has the data samples 5, 8, 9 and 15, a feature may order those as 9,8,15,5. Then, $\nu(1) = 9$, $\nu(2) = 8$, $\nu(3) = 15$ and $\nu(4) = 5$.

Note that there is a key difference regarding the indices notations. We write $i$ the absolute index of sample from the dataset $\mathcal{D}$ while $l$ refers to the ordered count of samples inside a specific node. The index $t$ is the absolute index according to the ordering $\nu(l)$.

We introduce 3 helping variables. The first one is the sample-wise cluster adversarial stocks:

$$\boldsymbol{\omega}_{k,i} = \sigma(\mathcal{C}_k \times \{x_t\}) \tag{39}$$

which does not depend on the ordering specified by a feature and will ease the computation of both $\sigma(\mathcal{S}_L \times \mathcal{C}_k)$ and $\sigma(\mathcal{S}_R \times \mathcal{C}_k)$. We can shortly write that $\boldsymbol{\omega} = \boldsymbol{CZ\kappa}$. To alleviate the computations of $\sigma(\mathcal{S}_L^2)$ and $\sigma(\mathcal{S}_R^2)$, we introduce the ordering-dependent variables:

$$\alpha_t^\nu = \sum_{\substack{l=1\cdots|\mathcal{N}_p| \\ t>\nu(l)}} \kappa_{\nu(l),t}, \tag{40}$$

and:

$$\beta_t^\nu = \sum_{\substack{l=1\cdots|\mathcal{N}_p| \\ t<\nu(l)}} \kappa_{\nu(l),t}. \tag{41}$$

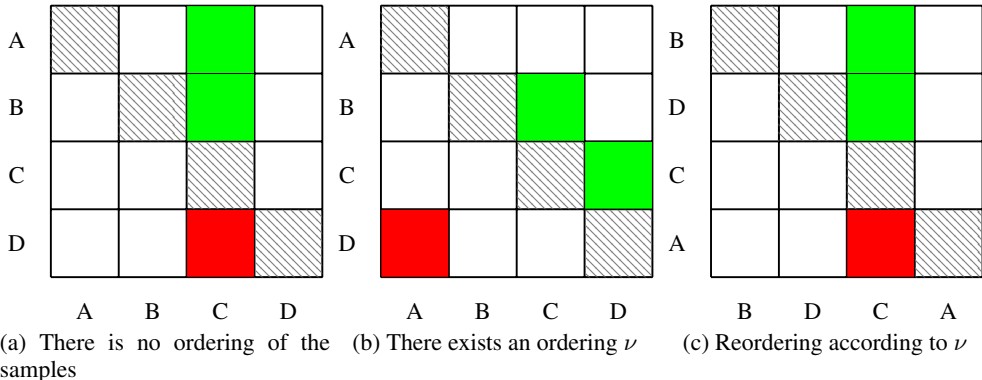

(a) There is no ordering of the samples    (b) There exists an ordering $\nu$    (c) Reordering according to $\nu$

Figure 5: An example of the value of the variables $\alpha_3$ and $\beta_3$ which respectively are the sum of green squares and red squares on the kernel matrix of the elements A, B, C and D in a leaf. In 5b and 5c, the ordering is $\nu(\{1,2,3,4\}) = \{B,D,C,A\}$.

These two variables verify a constant sum $\beta_t^\nu + \alpha_t^\nu + \kappa_{tt} = \sigma(\{x_t\} \times \mathcal{N}_p)$ for all $t \in \mathcal{N}_p$. We provide a visual intuition of the definition of these variables in Fig. 5. Once the variables $\omega_{k,i}$, $\alpha_t^\nu$, $\beta_t^\nu$ are initialised, we can compute all split gains with simple additions.

The initialisation of the variables is easy. For the split self-stock, we have:

$$\sigma(\mathcal{S}_L^0 \times \mathcal{S}_L^0) = 0, \tag{42}$$
$$\sigma(\mathcal{S}_R^0 \times \mathcal{S}_L^0) = \sigma(\mathcal{N}_p^2), \tag{43}$$

because starting from no sample yields all content of the node $\mathcal{N}_p$ to the right split $\mathcal{S}_R^0$. The adversarial stocks follow the same logic:

$$\sigma(\mathcal{S}_L^0 \times \mathcal{C}_k) = 0, \tag{44}$$
$$\sigma(\mathcal{S}_R^0 \times \mathcal{C}_k) = \sigma(\mathcal{N}_p \times \mathcal{C}_k), \tag{45}$$

where the last term is simply an element of $\boldsymbol{\gamma}$ indexed by the respective leaf and cluster. The iterations then consist in removing adequate adversarial stock or self-kernel stock:

$$\sigma(\mathcal{S}_L^l \times \mathcal{S}_L^l) = \sigma(\mathcal{S}_L^{l-1} \times \mathcal{S}_L^{l-1}) + 2\alpha_{\nu(l)}^\nu + \kappa_{\nu(l),\nu(l)}, \tag{46}$$

$$\sigma(\mathcal{S}_R^l \times \mathcal{S}_R^l) = \sigma(\mathcal{S}_R^{l-1} \times \mathcal{S}_R^{l-1}) - 2\beta_{\nu(l)}^\nu - \kappa_{\nu(l),\nu(l)}. \tag{47}$$

The adversarial scores are easier to update:

$$\sigma(\mathcal{S}_L^l \times \mathcal{C}_k) = \sigma(\mathcal{S}_L^{l-1} \times \mathcal{C}_k) + \boldsymbol{\omega}_{k,\nu(l)}, \tag{48}$$

and conversely:

$$\sigma(\mathcal{S}_R^l \times \mathcal{C}_k) = \sigma(\mathcal{S}_R^{l-1} \times \mathcal{C}_k) - \boldsymbol{\omega}_{k,\nu(l)}. \tag{49}$$

Thanks to these iterative variables, the iterative computation of all kernel stocks can be achieved in $\mathcal{O}(|\mathcal{N}_p|K)$ for a specific feature and node samples $\mathcal{N}_p$ instead of a naive $\mathcal{O}(|\mathcal{N}_p|^2 + n|\mathcal{N}_p|)$ as summarised in Algorithm 1. Notice that we omitted the pre-computing of $\alpha^\nu$ and $\beta^\nu$ which is done in $\mathcal{O}(|\mathcal{N}_p|^2)$. The pre-computing of $\boldsymbol{\omega}$ takes $\mathcal{O}(n^2)$ and can be done in advance at the tree-level.

Finally, we can optimise the computation of all splits.

### D.1.4 COMPLETE PICTURE

---
**Algorithm 2** Training SAGITTARIUS
---

**Require:** $\mathcal{D} = \{\boldsymbol{x}_i\}_{i=1}^n$ a dataset, $\boldsymbol{x}_i \in \mathbb{R}^d$
**Require:** $K_{\max} >= 2$ the maximum number of allowed clusters
**Require:** $d_{\max} \in \{1, d\}$ the maximum number of feature to consider per split.
**Require:** $L_{\max} \leq n$ the maximum number of leaves

1: **function** TRAINKAURI($\mathcal{D}, K_{\max}, d_{\max}, L_{\max}$)
2:     $\boldsymbol{\kappa} \leftarrow \langle \varphi(\mathcal{D}), \varphi(\mathcal{D}) \rangle$        ▷ Kernel value of samples, $n \times n$
3:     Initialise $\boldsymbol{Z}$ and $\boldsymbol{Y}$     ▷ All samples in one leaf, leaf belongs to only one cluster
4:     Initialise the tree structure in `Tree`.
5:     `Leaves` $\leftarrow$ `List(0)`        ▷ Only one starting leaf to explore
6:     $\bar{\Delta\mathcal{L}} \leftarrow \infty$        ▷ Last gain value
7:     `BestSplit` $\leftarrow \emptyset$        ▷ The best split proposal
8:     **while** `Leaves` $\neq \emptyset \wedge |$`Tree`$| \leq L_{\max} \wedge \bar{\Delta\mathcal{L}} > 0$ **do**
9:        $\Delta\mathcal{L} \leftarrow 0$        ▷ Best split achieved so far
10:        $\boldsymbol{\Lambda} \leftarrow \boldsymbol{Z\kappa}$        ▷ Compute $\sigma(\mathcal{N}_p \times \{j\})$
11:        $\boldsymbol{\omega} \leftarrow \boldsymbol{Y\Lambda}$        ▷ Compute $\sigma(\mathcal{C}_k \times \{j\})$
12:        $\boldsymbol{\gamma} \leftarrow \boldsymbol{\omega Z}^\top \boldsymbol{Y}^\top$        ▷ Compute $\sigma(\mathcal{C}_k \times \mathcal{C}_{k'})$
13:        $|\mathcal{C}_k| \leftarrow \boldsymbol{YZ1}_n$        ▷ Sizes of clusters
14:        **for** $p \in$ `Leaves` **do**
15:           $\mathcal{N}_p \leftarrow \{i|\boldsymbol{Z}_{ji} == 1\}$        ▷ Find the indices of leaf $p$
16:           $k \leftarrow \arg\max_{k'} \boldsymbol{Y}_{k',p}$        ▷ The current cluster of leaf $p$
17:           **for** $f \leftarrow 1$ to $d_{\max}$ **do**
18:              $\nu \leftarrow$ Argsort($\{\boldsymbol{x}_{if}|i \in \mathcal{N}_p\}$)
19:              $\tilde{\Delta\mathcal{L}}$, split $\leftarrow$ FINDBESTSPLIT($\mathcal{N}, j, \nu, \boldsymbol{\kappa}, \boldsymbol{\Lambda}, \boldsymbol{\omega}, \boldsymbol{\gamma}, |\mathcal{C}_k|, k$)
20:              **if** $\tilde{\Delta\mathcal{L}} > \Delta\mathcal{L}$ **then**
21:                 $\Delta\mathcal{L} \leftarrow \tilde{\Delta\mathcal{L}}$
22:                 `BestSplit` $\leftarrow$ split $\cup (p, f)$ ▷ Add node and feature information to the best split
23:              **end if**
24:           **end for**
25:        **end for**
26:        **if** $\Delta\mathcal{L} > 0$ **then**
27:           Remove the best leaf from the list `Leaves` and add the children of the split in `Leaves` if they satisfy structural constraints.
28:           Update `Tree` using `BestSplit`
29:           Update $\boldsymbol{Z}$ and $\boldsymbol{Y}$.
30:        **end if**
31:     **end while**
32:     **return** `Tree`
33: **end function**

---

The complete algorithm of Kauri is written in Algorithm 2. We estimate the complexity of the split search to $\mathcal{O}(n((L+d)(n+K)+dL)+L^2(d+K))$ at worst and $\mathcal{O}(n(n+K)(d+L)+KL^2)$ at best, where $L$ is the current number of leaves.

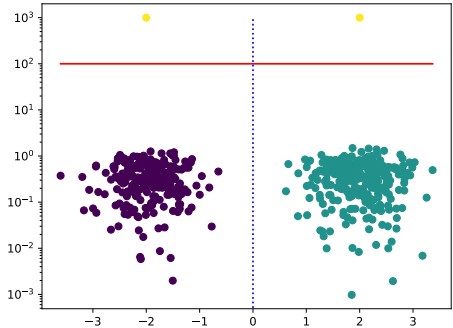

Figure 6: A dataset proposed by Moshkovitz et al. (2020) consisting in two isotropic Gaussian distributions and a cluster of two points distant on the y-axis. In order to split optimally the clusters, a decision tree should start with a y-axis split (solid red line) then use an x-axis split (dashed blue line) to separate the two Gaussian distributions.

## E    BENCHMARK PREPROCESSING AND HYPERPARAMETERS

We used standard scaling for all datasets. All categorical variables were one-hot-encoded, except for the US congressional votes dataset, where we encoded specifically the answer yes as 1, the no as -1 and the unknown votes as 0. In other datasets, we tossed away all samples that presented missing values.

All of our runs with Douglas were performed with 200 epochs for a learning rate of $10^{-3}$ with an Adam optimiser. We varied the batch size with 32 for the iris dataset, 64 for the rbeast cancer, haberman and wine datasets, 256 for avila and poker datasets and finally 128 for the remainder. In the case of the large datasets avila and poker, we downsized the number of epochs to 50 as the number of batches was great enough.

The Kauri was unconstrained on most dataset except avila and poker where we restrained the splits to occur on nodes that had at least 20 samples in order to speed up the training.

## F    RELATED MOTIVATING EXAMPLES

When trying to motivate their algorithm, Moshkovitz et al. (2020, Figure 2b) create a simple dataset where the combination of KMeans+Tree would solve the task with excellent accuracy, yet with non-optimal splits.

This dataset consists in 3 clusters. The first two ones are respectively drawn from $\mathcal{N}([2,0]^\top, \epsilon I_2)$ and $\mathcal{N}([-2,0], \epsilon I_2)$ with $\epsilon$ small enough. The last cluster contains two points located at $(-2, v)$ and $(2, v)$. We plot in Figure 6 a sample of such dataset for $v = 1000$.

A decision tree learning from KMeans labels will start by separating the samples along the x-axis. This non-optimal choice then requires two splits on the left- and right-hand sides to then separate the Gaussian distributions from the third cluster.

The optimal choice, achieved by ExKMC as well as Kauri, starts by cutting on the y-axis, separating thus all Gaussian distributions from the third cluster. A single split afterwards is sufficient for separating the two Gaussian distributions.

This shows that the explanation quality brought by Kauri can be of better quality for the same clustering results.

Table 6: Average ARI scores (std) over 30 runs on small datasets for the pure-NumPy implementation of the Douglas tree algorithm.

| Dataset | Breast cancer | Haberman | Iris | Votes | Vowel | Wine |
|---------|---------------|----------|------|-------|-------|------|
| ARI | 0.84 (0.03) | 0.00 (0.01) | 0.54 (0.09) | 0.02 (0.02) | 0.30 (0.15) | 0.17 (0.09) |

Table 7: ARI scores $_{std}$ (greater is better) of Kauri, Douglas and other methods after 30 runs on random subsamples of 80% of the input datasets. All models are limited to finding 4 times more leaves than clusters.

| Dataset | Kauri | KMeans+Tree | ExKMC |
|---------|-------|-------------|-------|
| Avila | $0.04_{0.00}$ | $0.06_{0.01}$ | $0.06_{0.01}$ |
| Cancer | $0.86_{0.02}$ | $0.84_{0.03}$ | $0.86_{0.02}$ |
| Car | $0.05_{0.06}$ | $0.07_{0.06}$ | $0.06_{0.07}$ |
| Congress | $0.50_{0.04}$ | $0.56_{0.04}$ | $0.53_{0.04}$ |
| Digits | $0.41_{0.03}$ | $0.45_{0.04}$ | $0.44_{0.04}$ |
| Haberman | $0.01_{0.05}$ | $0.00_{0.00}$ | $0.00_{0.01}$ |
| Iris | $0.61_{0.03}$ | $0.62_{0.04}$ | $0.61_{0.03}$ |
| Mice | $0.19_{0.02}$ | $0.18_{0.02}$ | $0.18_{0.01}$ |
| Vowel | $0.13_{0.04}$ | $0.15_{0.02}$ | $0.17_{0.02}$ |
| Wine | $0.89_{0.03}$ | $0.91_{0.03}$ | $0.90_{0.03}$ |

# G    PURE NUMPY DOUGLAS PERFORMANCES

We list here in Table 6 the performances of our own pure NumPy implementation of the Douglas algorithm. Overall, the results are slightly below the average performance of the Pytorch version of Douglas, except for the Wine dataset where we lost 0.3 points of ARI.

# H    PERFORMANCES WITH MORE LEAVES THAN CLUSTERS

We run here the exact same benchmark as proposed in section 5.1, except we seek to compare Kauri with the ExKMC method. To that end, all trees are now limited to 4 times more leaves than clusters following the result of Frost et al. (2020). Contrary to section 5.1, the excessive number of leaves will necessarily imply that multiple leaves might explain a single cluster. We chose then to measure the WAES because we want to emphasize more the complexity of the explanation of a cluster rather than the depth of trees. We report the ARI in Table 7 and the WAES in Table 8.

Table 8: WAES scores $_{std}$ (lower is better) of Kauri and related works after 30 runs on random subsamples of 80% of the input datasets. All models are limited to finding 4 times more leaves than clusters.

| Dataset | Kauri | KMeans+Tree | ExKMC |
|---------|-------|-------------|-------|
| Avila | $8.29_{0.32}$ | $\mathbf{6.42_{0.16}}$ | $10.77_{0.18}$ |
| Cancer | $\mathbf{2.49_{0.18}}$ | $2.77_{0.24}$ | $3.20_{0.33}$ |
| Car | $\mathbf{2.00_{0.00}}$ | $2.05_{0.11}$ | $2.05_{0.06}$ |
| Congress | $\mathbf{1.65_{0.33}}$ | $2.42_{0.24}$ | $2.65_{0.29}$ |
| Digits | $6.26_{0.21}$ | $\mathbf{5.74_{0.25}}$ | $9.40_{0.46}$ |
| Haberman | $\mathbf{1.08_{0.18}}$ | $2.03_{0.23}$ | $2.41_{0.47}$ |
| Iris | $\mathbf{2.49_{0.20}}$ | $2.59_{0.25}$ | $3.01_{0.26}$ |
| Mice | $5.53_{0.23}$ | $\mathbf{5.42_{0.31}}$ | $7.22_{0.40}$ |
| Vowel | $3.10_{0.20}$ | $\mathbf{2.85_{0.42}}$ | $3.61_{0.49}$ |
| Wine | $\mathbf{2.72_{0.38}}$ | $2.87_{0.31}$ | $3.53_{0.50}$ |

We observed in Table 7 that Kauri often performs on par with ExKMC regarding the clustering performances. Notably, these performances are close to the KMeans+Tree baseline, except for the US congressional votes dataset. We also find that we maintain scores with Kauri that are steadily lower than ExKMC regarding WAES in Table 8.

To further investigate some differences, we observe in Fig. 7 the differences in WAES scores between KMeans+Tree, Kauri and ExKMC (Frost et al., 2020). We observe that overall, for a fixed amount of leaves to use, we obtained explanations with lower WAES scores than ExKMC while maintaining an ARI that is close to KMeans+Tree.

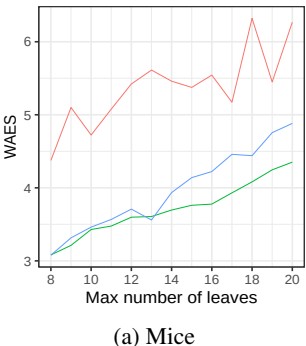

(a) Mice

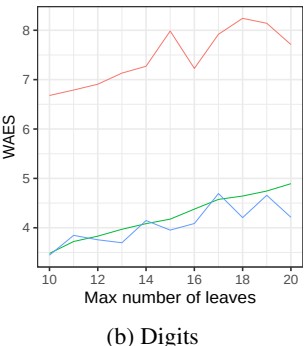

(b) Digits

Figure 7: Explanation scoring with WAES (lower is better) as the maximal number of leaves increases on the mice protein and digits datasets for Kauri (green), KMeans+Tree (blue) and ExKMC (red).

## I ON MODEL SELECTION

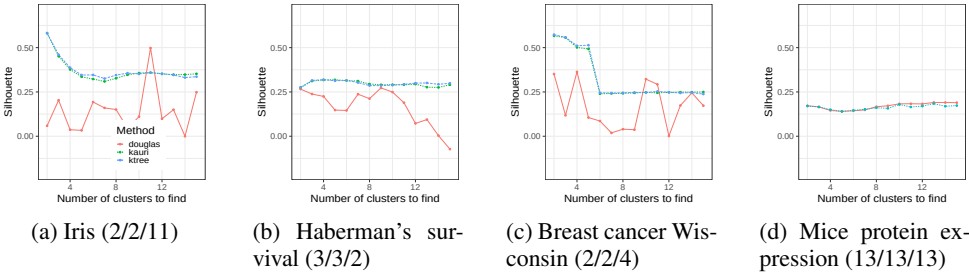

(a) Iris (2/2/11)

(b) Haberman's survival (3/3/2)

(c) Breast cancer Wisconsin (2/2/4)

(d) Mice protein expression (13/13/13)

Figure 8: Silhouette scores of Kauri and Douglas for various numbers of clusters compared with the KMeans+Tree algorithm. Selected number of clusters are written in parentheses as (Kauri / KMeans+Tree / Douglas).

One subsidiary question remains the choice of the number of clusters. In the general context of discriminative clustering, we cannot benefit from common Bayesian tools such as the Bayesian information criterion (Schwarz, 1978) or the integrated complete likelihood (Biernacki et al., 2000) because we cannot define a likelihood. Common tools for model selection with KMeans are the elbow method, despite recent critics (Schubert, 2023), the maximum silhouette score or the gap statistic (Tibshirani et al., 2001). The maximum silhouette consists in seeking the number of clusters with low intra-cluster distance and high inter-cluster distance. The gap statistic consists in comparing the progressive decrease of a weighted sum of squares against a null hypothesis and selecting the number of clusters at which the gap suddenly decreases beyond standard deviation bounds. We ran these two methods as examples on four datasets and summarised the results in the figures 8 and 9. We only computed the gap statistic for the Kauri method since running bootstrap estimates of the required weighted sum of squares for Douglas was too time-consuming. We compared the algorithms with the performances of the combination KMeans+Tree.

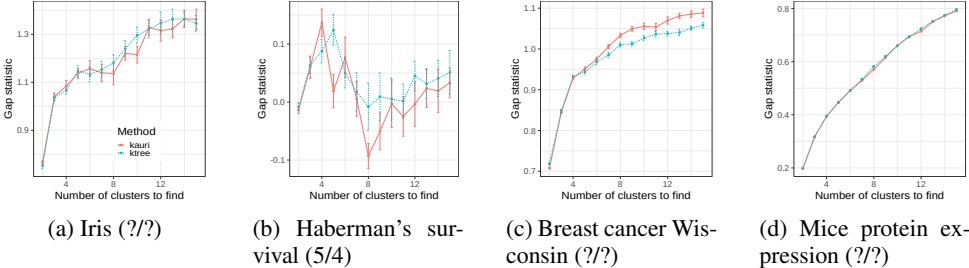

(a) Iris (?/?)  (b) Haberman's survival (5/4)  (c) Breast cancer Wisconsin (?/?)  (d) Mice protein expression (?/?)

Figure 9: Gap statistic curves of Kauri various numbers of clusters compared with the KMeans+Tree algorithm. The selected number of clusters are written in parentheses as (Kauri / KMeans+Tree). We wrote ? when we were unable to determine a relevant gap within the limit of one standard deviation.

To compute the gap statistic in model selection (Tibshirani et al., 2001), the central element to compute is the weighted sum of squares (WCSS) defined as:

$$W_K = \sum_{k=1}^{K} \frac{1}{2|\mathcal{C}_k|} \sum_{i,j \in \mathcal{C}_k} \|\boldsymbol{x}_i - \boldsymbol{x}_j\|_2^2. \tag{50}$$

A natural extension to compute this WCSS in the kernel KMeans is to switch from the usual Euclidean space to a Hilbert space $\mathcal{H}$. Then, we need to use the kernel trick to properly compute WCSS:

$$W_K = \sum_{k=1}^{K} \frac{1}{2|\mathcal{C}_k|} \sum_{i,j \in \mathcal{C}_k} \kappa(\boldsymbol{x}_i, \boldsymbol{x}_i) + \kappa(\boldsymbol{x}_j, \boldsymbol{x}_j) - 2\kappa(\boldsymbol{x}_i, \boldsymbol{x}_j). \tag{51}$$

However, that last equation is in fact equivalent to the subtraction of the data kernel and the objective function of Kauri as shown in the section B.2. We deduce:

$$W_K = \sigma(\mathcal{D} \times \mathcal{D}) - \mathcal{L}. \tag{52}$$

We can therefore use the gap statistic to select models with Kauri.

Our first main observation from both figures is that the Kauri algorithm follows very well the curves from the combination KMeans+Tree, therefore it seems reasonable to expect model selection methods to perform equally well with KMeans or Kauri in general. Our second observation is that apart from the mice protein dataset where all methods agreed with Silhouette scores to the same number of clusters (Fig. 8d), the number of clusters is often close or equal to the number of classes, except for Douglas on the iris dataset which Silhouette scores spiked at 11 clusters. However, the selection with the gap statistic was not as successful because we did not get any clear gap decrease apart from the Haberman's survival dataset (Fig. 9b) where the number of clusters is far from the number of classes. This is as well a good reminder that the number of classes is not necessarily the optimal number of clusters, an ill-defined concept.

