# OpenReview forum: "End-to-End Training of  Unsupervised Trees: KAURI and DOUGLAS"
_ICLR.cc/2024/Conference — Submitted to ICLR 2024_

### Official Review · Reviewer_nAXp · 2023-10-19

**Soundness:** 3 good
**Presentation:** 3 good
**Contribution:** 3 good
**Rating:** 3
**Confidence:** 3

**Summary:**

This paper introduces a framework for unsupervised tree-based end-to-end learning. This framework combines tree structures with generalized mutual information for clustering, resulting in two approaches: KAURI and DOUGLAS. KAURI focuses on maximizing a kernel-KMeans-like objective to iteratively create unsupervised splits by assigning tree leaves to either existing or new clusters. On the other hand, DOUGLAS harnesses the power of differential trees and the Wasserstein distance. KAURI is more suitable for small-scale datasets, while DOUGLAS excels with larger datasets that have fewer features.

**Strengths:**

The paper is readable and well-written. I found it practical to propose two algorithms that complement each other's weaknesses and can be mentioned for their respective suitable use cases. Furthermore, the paper takes into account not only the algorithms but also aspects such as fast implementation. It also includes considerations regarding computational cost estimation.

**Weaknesses:**

Several successful experimental cases are presented, yet the paper lacks theoretical backing. Although the proposed method is straightforward, it doesn't appear to offer a high degree of novelty. Consequently, the research's significance remains unclear.

The simplicity of the proposed method makes it particularly important to validate its effectiveness through numerical experiments. However, the descriptions of these experiments lack adequate detail. For instance:

- The DOUGLAS experiment is said to be limited by memory constraints, but there is no information about the specific memory requirements or the machine resources used.
- Performance metrics are mentioned, but the paper does not provide data on computational time and memory usage.
- The Appendix notes that the batch size for the DOUGLAS experiments varies depending on the dataset, but it does not explain the methodology behind this decision. A comparison of the amount of parameter tuning against a benchmark is also needed for further validation.
- In the Appendix, it's stated that the handling of categorical variables varies depending on the dataset. However, information is only provided for the US congressional votes dataset, affecting the experiment's reproducibility.

(Minor comment: The capitalization of "KAURI/Kauri" and "DOUGLAS/Douglas" is inconsistent, and there is a lack of consistency in notation.)

**Questions:**

1: The objective of KAURI is introduced as being equivalent to optimizing the K-means objective. In that case, what should we consider as the motivation behind this study? I would like to understand the rationale for using this research approach instead of traditional Kernel KMeans. While one example is provided in Appendix E, I also wondered if there might be cases where KAURI doesn't perform well conversely. I believe the clear difference lies in the fact that it is an end-to-end approach. What might be the motivation behind this choice?

2: How do the experimental results vary when the temperature parameter $\tau$ in Equation 9 is modified? Although the temperature is set to 0.1 throughout this paper, it is known that this parameter is crucial in the context of differentiable trees. (See Reference [1])

3: Please provide information on the machine resources used, computation time, memory usage, and the amount of parameter tuning (See Weakness part).

[1]: A Neural Tangent Kernel Perspective of Infinite Tree Ensembles, Kanoh&Sugiyama(2022), ICLR2022

---

> ### Author Response · Authors · 2023-11-17
>
> > The authors methods connect KMeans directly with tree-based models. This simplifies the clustering process if one would like to use a tree for this purpose. The connection with mutual information is natural.
>
> We thank you for your thorough review and appreciate your understanding of our work.
>
> > The objective of KAURI is introduced as being equivalent to optimizing the K-means objective. In that case, what should we consider as the motivation behind this study? I would like to understand the rationale for using this research approach instead of traditional Kernel KMeans. While one example is provided in Appendix E, I also wondered if there might be cases where KAURI doesn't perform well conversely. I believe the clear difference lies in the fact that it is an end-to-end approach. What might be the motivation behind this choice?
>
> The global motivation for the study was to introduce a complete framework on tree architecture and GEMINI training. The link with kernel KMeans is here to serve as a theoretical justification for the expected performances of one specific case of this framework: KAURI.
>
> We believe that there exists indeed a clear difference between the end-to-end training approach of such a tree compared to twofold methods, especially with Table 4 (now 5) displaying a shallower structure on average, Figure 2 highlighting how we deal better with non-axis aligned decision boundary beyond Appendix E (now F).
>
> Still, none of our experiments showed KAURI as significantly underperforming compared to kernel KMeans. In addition, the normalised KMeans scores suggested by reviewer #2 (dexj) that we added to the paper remain better or close on average.
>
> > How do the experimental results vary when the temperature parameter in Equation 9 is modified? Although the temperature is set to 0.1 throughout this paper, it is known that this parameter is crucial in the context of differentiable trees. (See Reference [1])
>
> Our preliminary study on the parameters did not reveal significant changes in ARI or KMeans score for temperatures ranging from 0.01 to 10. Therefore, we chose to let the default parameter to 0.1 as advised by Yang et al., authors of DNDTs, in our experiments.
>
> > Please provide information on the machine resources used, computation time, memory usage, and the amount of parameter tuning (See Weakness part).
>
>
> To begin with, the memory complexity of DOUGLAS is that of DNDTs. It grows in $\mathcal{O}((T+1)^d)$ with $d$ the number of features and $T$ the number of cuts, e.g. in section 5.1: 20 features cut in 2 is approx. 2^20 parameters. Then, the autodiff framework and the number of samples make this number grow linearly. Consequently, with the double encoding which is required for the Wasserstein GEMINI, the model associated to the car evaluation dataset would already take multiple GBs which saturated our GeForce RTX 2080 (8GB). The time performances of the model's complete training could range from a couple of seconds for the Haberman survival (fewest features and 2 clusters) to half an hour for the Avila dataset (20,000 samples and 12 clusters). The number of clusters intervenes on the complexity of the GEMINI computation by a $K^2$ factor.
>
> For KAURI, beyond the complexity per split which could be at worst of $\mathcal{O}(n^2[(L+d)(n+K)+dL]+L^2(d+K))$ as reported in App. C.1.4 worst for $n$ samples, $L$ leaves, $d$ features and $K$ clusters, practical numbers would be roughly less than a second for the iris dataset, and close to a few minutes for the Avila dataset.
>
> > In the Appendix, it's stated that the handling of categorical variables varies depending on the dataset. However, information is only provided for the US congressional votes dataset, affecting the experiment's reproducibility.
>
> The other method used for the categorical variables in other datasets is one hot-encoding. While this was briefly mentioned in Table 2 for the car dataset, we clarified it in Appendix D (now E).

---

### Official Review · Reviewer_4Lqz · 2023-10-31

**Soundness:** 3 good
**Presentation:** 2 fair
**Contribution:** 2 fair
**Rating:** 5
**Confidence:** 3

**Summary:**

The authors propose two methods for fitting tree models to unlabeled data to cluster the data. The first model uses binary splits while the second uses k-ary splits with a differentiable splitting function. The methods are compared against baseline methods on 10 data sets. Cluster quality is measure by adjusted rand index (ARI) and interpretability is measured using weighted average depth (WAD).

**Strengths:**

The authors methods connect KMeans directly with tree-based models. This simplifies the clustering process if one would like to use a tree for this purpose. The connection with mutual information is natural.

**Weaknesses:**

The authors results show some improvement over KMeans combined with a supervised tree. But the improvement is small and I'm not sure the improved interpretability is sufficient to strengthen the contribution enough. Interpretability is always a very thorny issue. It is ultimately and under-specified property and its value is in the eye of the beholder.

I agree that explain another clustering output using a decision tree leaves something to be desired in terms of elegance. But clusters are often used as a form of explanation, which raises the question why does one need to explain a clustering output to begin with?

Why are divisive or agglomerative methods not compared against? They can produce trees, albeit perhaps not with annotated internal nodes. Yet, WAD doesn't require such annotations, so they can be evaluated as the authors have done.

**Questions:**

Section 2 strays into topics that are somewhat out-of-place in the document. Discussing the advantages and nuances of the method overly much before the method has been introduced on a technical level is premature. Can portions of section 2 be moved into the discussion?

---

> ### Author Response · Authors · 2023-11-17
>
> > The authors methods connect KMeans directly with tree-based models. This simplifies the clustering process if one would like to use a tree for this purpose. The connection with mutual information is natural.
>
> We thank you for appreciating our proposed simplifications of clustering processes among trees and its "natural connection with mutual information".
>
> > The authors results show some improvement over KMeans combined with a supervised tree. But the improvement is small and I'm not sure the improved interpretability is sufficient to strengthen the contribution enough. Interpretability is always a very thorny issue. It is ultimately and under-specified property and its value is in the eye of the beholder.
> > I agree that explain another clustering output using a decision tree leaves something to be desired in terms of elegance. But clusters are often used as a form of explanation, which raises the question why does one need to explain a clustering output to begin with?
>
> We understand that the interpretability can be considered a "thorny issue".  In this sense and in order to clarify further the advantages of the proposed end-to-end approach, we added performances regarding the KMeans score of each algorithm, as requested by as well by reviewer #2 (dexj). You may find the table of new results in the global response showing (new Table 4) that not only are our trees shallower, but also well-performing on KMeans objective. Thus, the gain is not only on the subjective topic of interpretability but as well on KMeans performances.
>
> Overall, we remind that interpretability is to begin with a desired property in clustering. As mentionned in the paper *Interpretable clustering: an optimization approach* (Bertsimas et al.,  2020) highlighted by reviewer #1 (xtwY) above: "```[...] Clustering algorithms provide little insight into the rationale for cluster membership, limiting their interpretability.```"
>
> > Section 2 strays into topics that are somewhat out-of-place in the document. Discussing the advantages and nuances of the method overly much before the method has been introduced on a technical level is premature. Can portions of section 2 be moved into the discussion?
>
> We thank you for the suggestion of structure and propose to move the section 2.3 into the appendix, and use the remaining space to both present the performances with k-means loss as suggested by reviewer #2 (dexj), then incorporate further references from reviewer #1.

---

### Official Review · Reviewer_dexJ · 2023-10-31

**Soundness:** 2 fair
**Presentation:** 3 good
**Contribution:** 2 fair
**Rating:** 3
**Confidence:** 4

**Summary:**

The paper proposes two algorithms to learn decision trees for clustering problems. Instead of using reference clustering algorithm such as k-means as some form of supervision for tree learning, the paper instead tries to learn the clustering tree directly without any reference supervision. By adapting the recent work on a generalised mutual information (GEMINI) objective for clustering, the paper first proposes the algorithm KAURI to learn axis-aligned clustering trees in a greedy top-down induction way. The second algorithm (DOUGLAS) adapts differentiable Deep neural decision trees to optimize a variation of GEMINI (Wasserstein-GEMINI). Experiments on smaller-scale datasets are conducted and compared with 4 k-means-based tree clustering methods.

**Strengths:**

1) The paper is well-written on the active area of research on interpretable clustering.
2) The paper uses novel clustering objective for learning clustering trees.

**Weaknesses:**

1) The paper does not provide sufficient motivation for the use of a generalised mutual information (GEMINI) objective for clustering. While the original paper (Ohl et al., 2022) shows good results on unsupervised neural network training, it is still not clear what makes this objective well-suited for clustering problems, particularly with trees.
2) The paper has quite limited novelty. It adapts a recent clustering objective into the traditional CART-type greedy recursive partitioning algorithm to learn the axis-aligned tree (KAURI algorithm). And similarly with DOUGLAS algorithm, which just uses the existing differentiable deep neural decision trees.
3) The paper attempts to motivate for the end-to-end learning of clustering trees rather than using existing clustering algorithm such as k-means as reference. However, both theoretically and experimentally the advantage of end-to-end learning has not been clearly demonstrated.
4) The datasets used in experiments seem to be quite small. The largest contains 20k points in 10 dimensions. Having a dataset of at least MNIST-level size can help to show its scalability.
5) Adjusted rand index measure used to compare the clustering performance is questionable. As far as I understand, adjusted rand index uses ground-truth class labels but clustering is an unsupervised problem. Reporting both the k-means objective and GEMINI objective might help as they are the objective function being optimized.

**Questions:**

1) Why is this particular neural decision tree used for the DOUGLAS algorithm? How interpretable are these trees? Why not just regular soft decision tree?

---

> ### Author Response · Authors · 2023-11-17
>
> We thank you for appreciating the writing and our use of novel objective functions for discriminative clustering.
>
> > [...] While the original paper (Ohl et al., 2022) shows good results on unsupervised neural network training, it is still not clear what makes this objective well-suited for clustering problems, particularly with trees.
>
> To elaborate on the choice of this objective (GEMINI), there exist very few clustering objectives that can train discriminative models, i.e. without parametric assumptions on the data distribution $p(x)$. Most of these objectives are related to mutual information. Yet, Ohl et al. highlighted the superiority of GEMINI to MI in discriminative clustering. Moreover, the connection of this objective function to KMeans makes it a sensible objective for clustering. These reasons motivate the choice of GEMINI for training unsupervised trees.
>
> > The paper has quite limited novelty. It adapts a recent clustering objective into the traditional CART-type greedy recursive partitioning algorithm to learn the axis-aligned tree (KAURI algorithm). And similarly with DOUGLAS algorithm, which just uses the existing differentiable deep neural decision trees.
>
> We would like to emphasize that we showed here novel types of splits that are not part of the initial CART algorithm, e.g. completely reallocating samples to other or new clusters. We extended GEMINI to train non-differentiable models, contrary to its initial design by Ohl et al. To the best of our knowledge, we are the first to optimise the KMeans objective within a decision tree without prior warm starts, explaining better or similar performances on average.
>
> > The paper attempts to motivate for the end-to-end learning of clustering trees rather than using existing clustering algorithm such as k-means as reference. However, both theoretically and experimentally the advantage of end-to-end learning has not been clearly demonstrated.
>
> As KAURI is guided by a kernel KMeans objective, it seems reasonable to us that the performances cannot exceed that of kernel KMeans. Still, we showed shallower tree structures in Table 3, especially when the ideal decision boundary is not axis-aligned as illustrated by Fig. 2.
>
> > The datasets used in experiments seem to be quite small [...] Having a dataset of at least MNIST-level size can help to show its scalability.
> >
> > Adjusted rand index measure used to compare the clustering performance is questionable. As far as I understand, adjusted rand index uses ground-truth class labels but clustering is an unsupervised problem. Reporting both the k-means objective and GEMINI objective might help as they are the objective function being optimized.
>
> We already had an additional dataset (Poker hand dataset) containing approx 1 million entries. We originally discarded the dataset because all entries were 0 of score for ARI, but this experiment clearly shows that KAURI can scale to large datasets. We could not add DOUGLAS for this specific dataset due to the limitations of the Wasserstein GEMINI. We restored this dataset in the results, but discarded its ARI score because all models obtained 0.  Still, we would like to draw attention to App. C.1.4 where we report a complexity of $\mathcal{O}(n^2[(L+d)(n+K)+dL]+L^2(d+K))$ at worst for $n$ samples, $L$ leaves, $d$ features and $K$ clusters when choosing the optimal split.
>
> You may see the performance in the global response. As a follow-up on the performance metrics, we added the table with normalised KMeans score on the same benchmark as Table 3. The performances are written in the new Table 4.
>
> Specifically, we followed the suggestion of Reviewer #3 (4Lqz) regarding the placement of section 2.3 and relegated it to the appendix for the table above. This helped the incorporation of proposed related works by Reviewer #1 (xtwY) in the coherence of the section 2.
>
> >Why is this particular neural decision tree used for the DOUGLAS algorithm? How interpretable are these trees? Why not just regular soft decision tree?
>
> Assuming you refer to *Soft Decision Trees O. Irsoy, O. T. Yildiz, E. Alpaydin ICPR 21*, the differentiable neural tree allows us to have a ready-made full structure when training with the Wasserstein GEMINI. Specifically, we found that this objective is expensive to compute and the optimisation of one branch parameter after another like for soft decision trees combined with the combinatorial question of which cluster to affect each new branch would cost even more than the proposal of DNDTs architectures.
>
> The interpretability of DNDTs lies in their soft-binning functions which translate to affectation rules: certain combinations of bins lead to a specific class/cluster. That is why Yang et al., the authors, introduced the notion of *active cut points* that we used in section 5.3: these are the features for which the bins from the soft-thresholding actually leverage a partition of the data, rather than putting everything in a single bin.

---

### Official Review · Reviewer_xtwY · 2023-11-03

**Soundness:** 3 good
**Presentation:** 3 good
**Contribution:** 2 fair
**Rating:** 3
**Confidence:** 4

**Summary:**

The submission presents two algorithms for learning clustering trees. Both algorithms are guided by generalized mutual information and find axis-parallel splits. Results on UCI datasets show that the proposed approaches yield performance comparable to that obtained by using an existing two-stage process for finding clustering trees (k-means for labeling the data followed by CART).

**Strengths:**

Learning clustering trees is an interesting problem, and the proposed approach has an interesting connection to kernel k-means.

**Weaknesses:**

The performance of the proposed algorithms is comparable to the performance of the simple two-stage approach based on k-means and standard decision trees.

There is important work on clustering trees and density estimation trees that is not considered in the submission, see the references below:

Blockeel, H., Raedt, L. D., & Ramon, J. (1998, July). Top-Down Induction of Clustering Trees. In Proceedings of the Fifteenth International Conference on Machine Learning (pp. 55-63).

Fisher DH (1987) Knowledge acquisition via incremental conceptual clustering. Mach Learn 2(2):139–172

Ram P, Gray AG (2011) Density estimation trees. In: Proceedings of the 17th ACM SIGKDD International Conference on Knowledge Discovery and Data mining. ACM, pp 627–635

Bertsimas, D., Orfanoudaki, A., Wiberg, H.: Interpretable clustering: an optimization approach. Mach. Learn. 110(1), 89–138 (2021)

Gamlath, B., Jia, X., Polak, A., Svensson, O.: Nearly-tight and oblivious algorithms for explainable clustering. Adv. Neural. Inf. Process. Syst. 34, 28929–28939 (2021)

**Questions:**

N/A

---

> ### Author Response · Authors · 2023-11-17
>
> > Learning clustering trees is an interesting problem, and the proposed approach has an interesting connection to kernel k-means.
>
> We thank you for providing these interesting references and finding interesting the connection to kernel KMeans.
>
> > The performance of the proposed algorithms is comparable to the performance of the simple two-stage approach based on k-means and standard decision trees.
>
> We would like to draw attention to the fact that we maintain comparable clustering performances with shallower trees which we attribute to the end-to-end nature of training. Moreover, and with the questions of reviewer #2 (dexj), we added the table of KMeans score which highlights good or better scores on average for both KAURI and DOUGLAS.
>
> > There is important work on clustering trees and density estimation trees that is not considered in the submission, see the references below:
>
>
> As reviewer #3 (4Lqz) suggests a replacement of some parts of section 2 for clarity and relevance, we chose to move section 2.3 to appendices in order to get more room for the papers mentioned above.

---

> > ### Comment · Reviewer_xtwY · 2023-11-23
> > **Response to authors' comments**
> >
> > Thank you for your responses.
> >
> > I see that you have included two of the five references I have suggested. Regarding the three omissions, regarding Fisher's work and the density estimation trees, this is perhaps ok. However, it is difficult to see how Blockeel et al.'s clustering trees are not related to your work. In fact, I believe they should be included in your experiments, just like they are included in the experiments by Bertsimas et al.
> >
> > Regarding the new reference to Bertsimas et al., your paper has the following sentence now: "For example, Bertsimas et al. (2021) directly optimise the silhouette score, an internal clustering metric, yet report the need for warm start to train multivariate decision trees. " However, looking at their paper, the warm-start mechanism is used primarily to improve runtime (and, in some cases, improves clustering performance). Regardless, it seems important to include this method in your experiments as well (perhaps both with and without the warm-start mechanism).

---

### Author Response · Authors · 2023-11-17
**General response from authors**

We would like to thank the reviewers for their valuable feedback and very useful suggestions. We appreciate that you found interesting the connection with KMeans objective, the usage of novel clustering objectives and writing overall.

To address the concerns of all reviewers and satisfy suggestions, we fid the following:
+ We moved section 2.3 to now new Appendix A for improved clarity of related works
+ This granted sufficient room to add proposed relevant references in section 2.2
+ We included as well the normalised KMeans score performances to show how well the methods do (see table below)
+ Addition of a large dataset (Poker hand) with approximately 1 million samples, see table below
+ Minor spelling checks on methods names

We thank all reviewers for the suggested corrections.

For the normalised KMeans score:

| Dataset             | KAURI          | KMeans+Tree    | DOUGLAS        | EXSHALLOW      | RDM            | IMM            |
|:--------------------|:---------------|:---------------|:---------------|:---------------|:---------------|:---------------|
| Avila               | 1.22 (0.08) | 1.95 (0.07) | 1.72 (0.14) | 1.23 (0.10) | 1.30 (0.13) | **1.15 (0.07)** |
| Cancer       | 1.08 (0.02) | 1.08 (0.02) | **1.00 (0.01)** | 1.07 (0.02) | 1.31 (0.02) | 1.07 (0.01) |
| Car evaluation      | **1.00 (0.00)** | **1.00 (0.00)** | X            | **1.00 (0.00)** | 1.02 (0.03) | **1.00 (0.00)** |
| Congress | 1.05 (0.01) | 1.04 (0.01) | **1.00 (0.01)** | 1.04 (0.01) | 1.13 (0.02) | 1.04 (0.01) |
| Digits              | **1.13 (0.01)** | 1.19 (0.02) | X            | **1.13 (0.02)** | 1.24 (0.04) | **1.14 (0.02)** |
| Haberman   | **1.01 (0.00)** | **1.01 (0.00)** | 1.04 (0.02) | **1.01 (0.00)** | **1.01 (0.00)** | **1.01 (0.00)** |
| Iris                | **1.06 (0.04)** | **1.07 (0.04)** | 1.49 (0.24) | **1.06 (0.05)** | 1.29 (0.08) | **1.07 (0.05)** |
| Mice        | **1.05 (0.01)** | 1.09 (0.03) | X            | **1.05 (0.01)** | 1.33 (0.05) | 1.11 (0.03) |
| Poker          | **1.03 (0.00)** | 1.07 (0.02) | 1.16 (0.02) | 1.05 (0.00) | 1.07 (0.02) | 1.12 (0.05) |
| Vowel               | 1.06 (0.00) | 1.07 (0.01) | **1.04 (0.01)** | 1.07 (0.01) | 1.09 (0.01) | 1.09 (0.01) |
| Wine                | 1.09 (0.05) | 1.13 (0.05) | 1.11 (0.09) | **1.05 (0.02)** | 1.33 (0.05) | 1.05 (0.03) |

For the Poker hand dataset:

|    Score     | KAURI |     KMeans+Tree     |      DOUGLAS       |     ExShallow      |         RDM         | IMM |
|:------------|:----------------------|:-------------------|:------------------|:------------------|:-------------------|:---|
|     ARI (greater is better)     |         0 (0)               |       0 (0)              |           0 (0)         |           0 (0)         |           0 (0)          |  0 (0)   |
|     WAD (lower is better)     |        **3.26 (0)**                |        **3.26 (0)**             |        X            |          3.38 (0.05)          |       **3.28 (0.11)**              |  4.40 (0.45)   |

---

### Meta-Review · Area_Chair_ipR4 · 2023-12-08

**Metareview:**

All 4 reviewers recommend reject, based on a number of reasons. Please refer to the reviews. Most importantly, limited novelty and results not much better than the simple two-step approach of fitting a classification tree to the k-means clusters.

Besides the references from the reviewers, another highly relevant one is: "Optimal interpretable clustering using oblique decision trees", KDD 2022. This directly optimizes the k-means objective (or a general clustering objective) over a tree, and it uses sparse oblique trees, which are better suited than axis-aligned trees (since the k-means regions are polytopes).

**Justification For Why Not Higher Score:**

See metareview.

**Justification For Why Not Lower Score:**

N/A

---

### Decision · Program_Chairs · 2024-01-16

Reject